# Evolution-Aware Positive-Unlabeled Learning for Protein Design

## Abstract

We consider prediction of protein function, focusing on protein functionalities that enhance survival for one or more organisms. Sequencing these organisms provides plentiful positive training examples editdue to survivorship bias. In contrast, synthesizing and characterizing a protein with a mutation unseen in nature requires time-consuming wet lab experiments, making negative training examples scarce. Thus, datasets are often imbalanced, hindering classifier accuracy outside the training data. Positive-unlabeled (PU) learning attempts to address this issue by considering unlabeled protein sequences to be part of the data and modeling them as positive with a probability called the class prior. This class prior is often constant. Our insight is that an understanding of evolution suggests a novel sequence-dependent class prior when learning from sequencing data. We propose Evo-PU, a PU learning framework that integrates our novel class prior to create a likelihood for training classifiers. We evaluate Evo-PU on multiple real-world tasks on influenza hemagglutinin protein. Using influenza genomic surveillance data and held-out laboratory assays of mutants unseen in nature, Evo-PU outperforms state-of-the-art PU learning, one-class classification (OCC), and deep generative model-based methods (DGM) on these real-world problems, demonstrating the benefit of combining evolutionary modeling with data-driven learning for protein design. We further assess Evo-PU on standard ProteinGym benchmarks, focusing on protein overall fitness prediction. Evo-PU outperforms existing PU-learning and OCC baselines, while remains competitive to DGM-based approaches.

## 1 Introduction

A fundamental challenge in protein design is accurately classifying amino acid sequences according to whether they possess a particular biochemical functionality. Sequencing living organisms provides many examples of functional sequences, particularly when the functionality confers an evolutionary advantage – a phenomenon described as survivorship bias (Bermúdez-Guzmán et al., 2020; Thomas et al., 2022). As examples, mussel foot proteins are promising adhesives (Kord Forooshani & Lee, 2017), proteins from polar fish are promising for cryogenic storage of tissues (He et al., 2018), and viral fusion proteins are promising drug delivery vehicles (Brown et al., 2024). Evolutionary pressure guides natural selection to discover functional proteins that would be otherwise unlikely to arise and widespread sequencing gives researchers access to these sequences. However, the resulting datasets are imbalanced: non-functional protein mutants are absent because they inhibit survival and some functional sequences may also be missing. When the functionality of interest is essential for survival, all sequenced proteins carry a positive label. This imbalance complicates training accurate classifiers.

Positive-unlabeled (PU) learning (Liu et al., 2003; Bekker & Davis, 2020) provides a natural framework for training classifiers when only a subset of positive examples is labeled, there are no labeled negative examples, and the rest of the data is unlabeled. The unlabeled set may contain both positive and negative instances, but their labels are unknown. In protein design, sequenced functional proteins can be treated as labeled positives, while all other proteins of interest form an unlabeled set. Within this framework, Protein-PU (Song et al., 2021) is a specialized PU method developed for protein design using deep mutational scanning (DMS) data, where mutants are generated in the laboratory, typically one amino acid away from a parental sequence. Protein-PU models a functional protein detection assay as a labeling process in which all functional sequences have an equal chance of being labeled positive (the "detection probability"). While effective for DMS, this assumption does not hold

for naturally-occurring sequences: evolutionary selection and mutational accessibility make some sequences much less likely to be detected, especially those distant from common natural variants. Furthermore, standard PU methods not designed for protein design (Bekker & Davis, 2020) often identify reliable negatives based on their similarity to known positives and use them, along with the positives, to train a classifier. Such approaches are ill-suited here, because even a few mutations can make a protein non-functional.

Alternative approaches to prediction with positive-only protein data include one-class classification (OCC) (Tax & Duin, 2001; Perera et al., 2021), which trains only on positive sequences to identify outliers, and deep generative models (DGMs) trained on multiple sequence alignments (MSA) to capture evolutionary conservation and estimate functional likelihood (Meier et al., 2021; Frazer et al., 2021; Thadani et al., 2023). While DGMs effectively predict the overall fitness score of whole protein sequences, they are challenged to capture the exact functional nuances of short, locally acting peptide motifs within these sequences, which are more relevant for domain scientists to understand and design protein functions for applications such as disease surveillance and therapeutic developments. A detailed review of existing methods aligned with these approaches is provided in Appendix A.

To overcome the limitations of PU-learning, OCC, and deep generative model-based approaches for protein design, we model protein evolution within a PU learning approach and develop a novel *sequence-dependent* detection probability model. Our essential insight is that functional nucleotide sequences close to sequences prevalent in nature are more likely to be detected than functional sequences far away. This is because a nearby sequence has a higher chance of being formed by mutation from an existing sequence. Using this detection probability model, we propose a novel framework for PU learning from sequencing of naturally occurring nucleotide sequences. We demonstrate Evo-PU's effectiveness on three real-world tasks: screening viral epitopes for immune evasion, identifying peptides with binding specificity to a human receptor and classifying peptides with viral fusion activity. Using influenza genomic surveillance data and held-out laboratory assays of mutants not observed in nature, Evo-PU achieves superior performance compared to state-of-the-art PU learning, OCC, and DGM-based methods. We further evaluate Evo-PU on standard *ProteinGym* benchmarks (Notin et al., 2023), which focus on predicting overall fitness rather than specific biochemical properties, and find that Evo-PU improves upon PU-learning and OCC baselines, while DGMs that designed for fitness prediction perform best in this regime.

## 2 METHOD

In this section, we present the Evo-PU framework, where protein sequences are modeled via a data-generating process inspired by natural selection. This process defines the likelihood of observing sequences based on functionality and the emergence of their nucleotide variants. Exact computation is infeasible due to the large sequence space, so we use approximations that focus on biologically plausible sequences. The resulting likelihood is then used to train a probabilistic classifier. We also compare Evo-PU to classic binary and Protein-PU likelihoods, showing how it generalizes them via sequence-dependent observation probabilities, and highlight key distinctions from DGM-based methods.

### 2.1 NATURAL SELECTION AS A DATA-GENERATING PROCESS

Proteins are sequences of amino acids that perform functional tasks in living organisms. Let $\mathcal{A}$ denote the set of 20 natural amino acids, and consider proteins of length $L$, so that the set of all amino acid sequences is $\mathcal{X} = \mathcal{A}^L$. Each amino acid is encoded by a codon of three nucleotides drawn from $\mathcal{N}$ (either $\{A, C, G, T\}$ for DNA or $\{A, C, G, U\}$ for RNA). Some codons are stop signals and do not encode amino acids; we define the set of valid nucleotide sequences of length $3L$ as $\mathcal{Y} \subset \mathcal{N}^{3L}$, with each $y \in \mathcal{Y}$ translating deterministically to $x \in \mathcal{X}$ via $T : \mathcal{Y} \to \mathcal{X}$. For any $x \in \mathcal{X}$, we denote $\mathcal{Y}(x) = T^{-1}(x) \subset \mathcal{Y}$ as the set of all nucleotide sequences encoding $x$.

Each amino acid sequence $x$ is associated with a binary variable $A(x)$ indicating whether the corresponding protein exhibits a property of interest (e.g., immune evasion, bind to host cell membrane receptor or fuse with host cell membrane). Specifically, $A(x) = 1$ if the protein exhibits the property and 0 otherwise. In Evo-PU, we model $\mathbb{P}(A(x) = 1)$ via a parametrized function $p_a(x; \theta)$, where

$\theta \in \Theta$ is a model parameter to be estimated by maximum likelihood estimation (MLE) along with other nuisance parameters.

To model sequence observation, we consider substitution-only evolution at the nucleotide level, so sequences remain length $3L$ and their corresponding proteins length $L$. Let $E(y)$ be a binary variable indicating whether nucleotide sequence $y$ has emerged. By "emergence", we mean that mutation has created the nucleotide sequence in an organism. We model the emergence probability of $y$ using a parametrized function $p_e(y; \alpha)$ detailed in Section 2.4, so that $\mathbb{P}(E(y) = 1) = p_e(y; \alpha)$. In our framework, we treat $\alpha$ as a nuisance parameter to be estimated via MLE.

Since we consider the properties of such proteins to be crucial for an organism's survival and ultimately subject to evolutionary surveillance, it is *necessary* for a nucleotide sequence $y$ to be observed that (1) $y$ has emerged ($E(y) = 1$) and (2) the corresponding protein $T(y)$ possesses the property ($A(T(y)) = 1$). Observation is not guaranteed even if these conditions hold, due to sampling limitations or ecological prevalence. We capture this using a nucleotide-level observability variable $O_{\mathcal{Y}}(y)$, which is 1 if the sequence $y$ is detected and 0 otherwise. Formally, we model

$$O_{\mathcal{Y}}(y) \mid A(T(y)), E(y) \sim \text{Bernoulli}(q(y)),$$

with

$$q(y) = \begin{cases} p_o(y), & E(y) = 1 \text{ and } A(T(y)) = 1, \\ 0, & \text{otherwise.} \end{cases}$$

For simplicity, we often take $p_o(y) = p_o$ constant across all sequences, although it could be sequence-specific in general. We treat $p_o$ as a nuisance parameter to be estimated via MLE, together with $\theta$ and $\alpha$. This definition captures the fact that an amino acid sequence can only be observed in the dataset if at least one of its corresponding nucleotide sequences is detected. Define $O_{\mathcal{X}}(x)$ to be the observability binary random variable at amino acid level. As a direct consequence of this model, any observed amino acid sequence must be functional: $O_{\mathcal{X}}(x) = 1 \implies A(x) = 1$.

Finally, under model parameters $\theta$, $p_o$, and $\alpha$, we assume that $\{A(x) : x \in \mathcal{X}\}$ are mutually independent, $\{E(y) : y \in \mathcal{Y}\} \cup \{O_{\mathcal{Y}}(y) : y \in \mathcal{Y}\}$ are conditionally independent given $\{A(x) : x \in \mathcal{X}\}$. As a result, the amino acid–level observabilities, $\{O_{\mathcal{X}}(x)\}$ are independent across $x$, which allows the likelihood of observing amino acid sequences to factorize into a product form, as we will show in the next section.

## 2.2 BIOLOGICALLY-REALISTIC LIKELIHOOD FUNCTION

Using the model from Section 2.1, we can express the probability of observing an amino acid sequence $x$. While it is possible to define observability at the nucleotide level, we focus on amino acid–level observability due to its smaller, more tractable search space. Nonetheless, the framework can be generalized to nucleotide sequences.

Since possessing functional of interest is the prerequisite to be surveilled in tasks discussed in this work, we have

$$\mathbb{P}(O_{\mathcal{X}}(x) = 1) = \mathbb{P}(O_{\mathcal{X}}(x) = 1 \mid A(x) = 1)\,\mathbb{P}(A(x) = 1).$$

Conditional on $A(x) = 1$, the nucleotide observabilities $\{O_{\mathcal{Y}}(y) : y \in \mathcal{Y}(x)\}$ are independent. Therefore, this gives

$$\begin{aligned}
\mathbb{P}(O_{\mathcal{X}}(x) = 1 \mid A(x) = 1) &= \mathbb{P}(\exists y \in \mathcal{Y}(x) : O_{\mathcal{Y}}(y) = 1 \mid A(x) = 1) \\
&= 1 - \mathbb{P}(\forall y \in \mathcal{Y}(x) : O_{\mathcal{Y}}(y) = 0 \mid A(x) = 1) \\
&= 1 - \prod_{y \in \mathcal{Y}(x)} \left(1 - \mathbb{P}(O_{\mathcal{Y}}(y) = 1 \mid E(y) = 1, A(x) = 1)\mathbb{P}(E(y) = 1)\right) \\
&= 1 - \prod_{y \in \mathcal{Y}(x)} \left(1 - p_o p_e(y; \alpha)\right).
\end{aligned}$$

Thus, the marginal probability of observing amino acid sequence $x$ is

$$\mathbb{P}(O_{\mathcal{X}}(x) = 1) = p_a(x; \theta)\Big[1 - \prod_{y \in \mathcal{Y}(x)} \left(1 - p_o p_e(y; \alpha)\right)\Big].$$

Since the amino acid–level observabilities $\{O_{\mathcal{X}}(x)\}$ are independent across $x$, the likelihood of a dataset can be written as a product over sequences. Given observed amino acids $\mathcal{D}_n \subseteq \mathcal{X}$ and its complement $\mathcal{D}'_n = \mathcal{X} \setminus \mathcal{D}_n$, the log-likelihood therefore becomes

$$
\begin{aligned}
\ell(\theta, \alpha, p_o; \mathcal{D}_n) &= \sum_{x \in \mathcal{D}_n} \log \mathbb{P}(O_{\mathcal{X}}(x) = 1) + \sum_{x' \in \mathcal{D}'_n} \log \mathbb{P}(O_{\mathcal{X}}(x') = 0) \\
&= \sum_{x \in \mathcal{D}_n} \Big[ \log p_a(x; \theta) + \log \Big( 1 - \prod_{y \in \mathcal{Y}(x)} (1 - p_o p_e(y; \alpha)) \Big) \Big] \\
&\quad + \sum_{x' \in \mathcal{D}'_n} \log \Big[ 1 - p_a(x'; \theta)\big(1 - \prod_{y' \in \mathcal{Y}(x')} (1 - p_o p_e(y'; \alpha))\big) \Big].
\end{aligned} \tag{1}
$$

This log-likelihood can then be maximized to estimate $\theta$, $\alpha$, and $p_o$. Evo-PU maximizes this objective to train a probabilistic classifier.

## 2.3 COMPARISON OF EVO-PU WITH PU-LEARNING AND DGM-BASED METHODS

In this section, we compare our Evo-PU likelihood in Eq. 1 to the two existing PU-learning likelihood formulations and discuss the central distinctions between Evo-PU and DGM-based methods.

**Comparison to exiting PU-learning likelihoods**

Here we present the two related likelihood formulations within the PU-learning framework:

- **Classical binary classifier** likelihood, which assumes unobserved sequences lack the chemical property:

$$
\sum_{x \in \mathcal{D}_n} \log p_a(x; \theta) + \sum_{x' \in \mathcal{D}'_n} \log(1 - p_a(x'; \theta)); \tag{2}
$$

- **Protein-PU** likelihood proposed by (Song et al., 2021), which incorporates a fixed labeling efficiency parameter $q \in (0, 1)$:

$$
\sum_{x \in \mathcal{D}_n} \log q p_a(x; \theta) + \sum_{x' \in \mathcal{D}'_n} \log(1 - q p_a(x'; \theta)). \tag{3}
$$

All three likelihoods share a similar structure: a sum over observed sequences in $\mathcal{D}_n$, which are treated as positives in the classical framework or labeled in the Protein-PU framework, and a second sum over sequences not in $\mathcal{D}_n$.

The classical likelihood can be viewed as a special case of both Evo-PU and Protein-PU. Specifically, setting $p_o p_e(y; \alpha) = 1$ for all $y \in \mathcal{Y}(x)$ in Eq. 1 implies that every functional amino acid sequence is always observed whenever it exists, reducing Evo-PU to the classical likelihood in Eq. 2. Likewise, setting $q = 1$ in the Protein-PU likelihood in Eq. 3 means that every functional sequence is always labeled, which also recovers the classical form.

Comparing Evo-PU and Protein-PU directly, Protein-PU models labeling efficiency or class prior with a constant parameter $q$, representing the probability that a functional sequence is labeled. In contrast, Evo-PU models class prior as sequence-dependent through the term: $1 - \prod_{y \in \mathcal{Y}(x)} (1 - p_o p_e(y; \alpha))$, which reflects how likely a sequence is to emerge through mutational processes. This sequence-dependent formulation captures variability in the observation process that cannot be explained by a fixed efficiency parameter. As we will show in our numerical studies, this leads to better alignment with the natural data-generating process and improved predictive performance compared to approaches with constant labeling efficiency.

**Distinctions between Evo-PU and DGM-based methods**

We highlight here the key distinctions between Evo-PU and DGM-based approaches. First, DGM-based methods primarily capture patterns associated with overall evolutionary fitness, whereas Evo-PU is designed to identify sequence features that control a specific biochemical property essential for organismal survival. Second, Evo-PU bases its predictions on modeling why sequences are observed or missing, not solely on the distribution of sequences that appear in databases. While

DGMs infer constraints from observed sequences, they do not model the evolutionary forces and surveillance processes that determine the presence or absence of variants. By incorporating a sequence-dependent class prior that reflects mutational accessibility, Evo-PU directly leverages the evolutionary mechanism underlying sequence observability, offering a principled way to exploit this information for property-specific protein-function prediction.

## 2.4 NUCLEOTIDE EMERGENCE PROBABILITY MODEL

We now describe the nucleotide emergence probability $p_e(y; \alpha)$, a key component of the likelihood in Eq. 1. Let $\mathcal{D}_\mathcal{N}$ denote the set of observed nucleotide sequences at the current step, and $\mathcal{D}_\mathcal{N}^p$ the set of sequences that emerged naturally in the previous step (see Section 3.3 for details on estimating $\mathcal{D}_\mathcal{N}^p$).

Between steps, evolution occurs at the nucleotide level. For each sequence $y^p \in \mathcal{D}_\mathcal{N}^p$, let $c(y^p)$ denote its (unobserved) total count in the previous step, and let $\mathbb{P}(y^p \to y')$ be the probability that a host carrying $y^p$ transmits a mutated sequence $y'$ to the next host. Because mutations must overcome within-host bottlenecks to establish dominance (Petrova & Russell, 2018), we introduce a parameter $\alpha \in (0, 1)$ representing the probability that a mutated sequence successfully establishes dominance within the host.

For an unobserved sequence $y' \notin \mathcal{D}_\mathcal{N}$, we model the emergence probability as

$$p_e(y'; \alpha) = 1 - \prod_{y^p \in \mathcal{D}_\mathcal{N}^p} \left(1 - \mathbb{P}(y^p \to y') \, \alpha\right)^{c(y^p)}. \tag{4}$$

Here, $\left(1 - \mathbb{P}(y^p \to y') \, \alpha\right)^{c(y^p)}$ represents the probability that $y'$ fails to emerge from all replications of $y^p$. Assuming independent mutation events across replications, the product gives the probability that $y'$ fails to emerge from any sequence in the previous step, and its complement gives the probability that $y'$ emerges at least once.

In practice, both $\mathbb{P}(y^p \to y')$ and $\alpha$ are typically small as the mutation rate is low, while the counts $c(y^p)$ are large. Using the classical approximation $(1 + x)^a \approx e^{ax}$ for $|x| \ll 1$ and $|ax| \gg 1$, we obtain

$$p_e(y'; \alpha) \approx 1 - \exp\left(-\sum_{y^p \in \mathcal{D}_\mathcal{N}^p} \mathbb{P}(y^p \to y') \, \alpha \, c(y^p)\right). \tag{5}$$

For sequences already observed in the current step, $y \in \mathcal{D}_\mathcal{N}$, we set $p_e(y; \alpha) = 1$, since their emergence is already established.

## 2.5 ESTIMATING THE LIKELIHOOD FUNCTION VIA PROTEIN EVOLUTION

Evaluating the likelihood in Eq. 1 requires summing over all unobserved amino acid sequences $x'$ of length $L$ in $\mathcal{D}_n'$, the full set of sequences not present in $\mathcal{D}_n$. This set is exponentially large and makes exact computation of the likelihood intractable. Moreover, for each amino acid sequence (both observed and unobserved) $x$, evaluating the sequence-dependent class prior also requires a product over $\mathcal{Y}(x)$, a set of all nucleotide sequences that can translate to $x$, which can also be large for long amino acid sequences. Since in the reality mutations are rare, most of nucleotide sequences in $\mathcal{Y}(x)$ that have not been observed in the nucleotide dataset $\mathcal{D}_\mathcal{N}$ have negligible emergence probability $p_e(y; \alpha) \approx 0$, contributing minor to the likelihood.

To reduce computation, we approximate the likelihood by considering a smaller subset of amino acid sequences $\hat{\mathcal{D}}_n' \subset \mathcal{D}_n'$, generated by the observed nucleotide sequence dataset $\mathcal{D}_\mathcal{N}$, and containing only sequences likely to emerge naturally, yet unobserved. Specifically, we construct $\hat{\mathcal{D}}_n'$ by first generating a set of unobserved nucleotide sequences $\hat{\mathcal{D}}_\mathcal{N}$ that contains nucleotide sequences with one point mutation away from any observed nucleotide sequence in observed set $\mathcal{D}_\mathcal{N}$. In $\hat{\mathcal{D}}_\mathcal{N}$, we include only those unobserved nucleotide sequences whose emergence probability $p_e(y'; \alpha) > \epsilon$ for some fixed $\epsilon, \alpha > 0$. Then, we construct $\hat{\mathcal{D}}_n' = \{T(y') \in \mathcal{X} : y' \in \hat{\mathcal{D}}_\mathcal{N} \text{ and } T(y') \notin \mathcal{D}_n\}$.

Moreover, to further reduce the computation in the term for class prior $\prod_{y \in \mathcal{Y}(x)} (1 - p_o p_e(y; \alpha))$, we restrict $\mathcal{Y}(x)$ for any $x \in \mathcal{D}_n \cup \hat{\mathcal{D}}_n'$ to $\hat{\mathcal{Y}}(x) = \mathcal{Y}(x) \cap (\mathcal{D}_\mathcal{N} \cup \hat{\mathcal{D}}_\mathcal{N}')$.

By replacing $\mathcal{D}'_n$ with the subset $\hat{\mathcal{D}}'_n$ and $\mathcal{Y}(x)$ with $\hat{\mathcal{Y}}(x)$ in Eq. 1, and using the emergence probabilities $p_e(y; \alpha) = 1, \forall y \in \mathcal{D}_\mathcal{N}$ and $p_e(y'; \alpha)$ as defined in Eq. 5 for all $y' \in \hat{\mathcal{D}}'_\mathcal{N}$, we approximate the log-likelihood function by:

$$
\ell_n(\theta, p_o, \alpha; \mathcal{D}_n) \approx \sum_{x \in \mathcal{D}_n} \Big[ \log p_a(x; \theta) + \log \Big( 1 - \prod_{y \in \hat{\mathcal{Y}}(x)} (1 - p_o p_e(y; \alpha)) \Big) \Big]
$$
$$
+ \sum_{x' \in \mathcal{D}'_n} \log \Big[ 1 - p_a(x'; \theta) \big( 1 - \prod_{y' \in \hat{\mathcal{Y}}(x')} (1 - p_o p_e(y'; \alpha)) \big) \Big] := \hat{\ell}_n(\theta, p_o, \alpha; \mathcal{D}_n).
$$
(6)

We then train the probabilistic classifier $p_a(x; \theta)$ by jointly estimating the classifier parameters $\theta$ and two nuisance parameters: nucleotide observability efficiency $p_o$, and the probability that an emerged sequence becomes dominant $\alpha$ by minimizing the loss function defined as the negative of this approximated log-likelihood:

$$
(\theta^*, p_o^*, \alpha^*) \in \underset{(\theta, p_o, \alpha) \in \Theta \times (0,1) \times (0,1)}{\arg\min} -\hat{\ell}_n(\theta, p_o, \alpha; \mathcal{D}_n).
$$
(7)

# 3 NUMERICAL EXPERIMENTS

Here, we present numerical experiments on both real-world influenza datasets and standard ProteinGym (Notin et al., 2023) benchmarks to evaluate the performance of Evo-PU. These two sets of tasks serve distinct purposes. The influenza tasks focus on identifying functional mutations in specific peptide motifs for downstream applications-precisely what Evo-PU is designed for-whereas the ProteinGym benchmarks assess model's ability to predict overall protein fitness, a broader objective.

## 3.1 PREDICTING FUNCTIONAL MOTIFS OF THE INFLUENZA HEMAGGLUTININ PROTEIN

**Problem background**: Influenza causes over 500,000 deaths worldwide each year (Stöhr, 2002; Thompson et al., 2009; Nair et al., 2011). Three critical drivers of influenza virus's ability to infect hosts are its ability to evade the human immune system, to bind with host cells, and to fuse with host cell membrane, all mediated by the functions of peptide domains located at different regions within its viral protein hemagglutinin (Epand, 2003). The Ca1 epitope (a combined of 11 amino acid residues) is one of five antigenic sites located in H1 subtype hemagglutinin (positions 169–173 and 206–208 and 238-240 per H3 numbering) (Wu & Wilson, 2017; Sriwilaijaroen & Suzuki, 2012), and it often mutates to escape the recognition of human immune response. Upon successful evasion of immune system, influenza binds with host cell via the binding domain in hemmaglutin (positions 134-138, 186-195 and 221-228 per H3 numbering) (Wiley et al., 1981; Yang et al., 2007), here in this study we combine the 23 amino acid residues and refer them as binding peptide. Upon binding, influenza can fuse with host cell membrane via hemagglutinin fusion domain, a consecutive 23-amino-acid-sequence named fusion peptide (Wiley & Skehel, 1987; Luo, 2012). Our goal for these experiments is to test the ability of Evo-PU in predicting mutants with vastly different functions, i.e. immune evasion, human receptor binding and membrane fusion.

**Dataset:** We obtained the prevalence data on host-infecting hemagglutinin protein nucleotide sequences collected between year 2001 and year 2024 (NCBI, 2024; Shu & McCauley, 2017). We extracted 7,383 unique nucleotide sequences that encode 504 unique amino acid sequences for fusion peptide mutants. In the binding peptide case, only human-infecting hemagglutinin protein nucleotide sequences were used, since different hemmaglutinin subtypes can bind with non-human hosts via their affinities with other types of influenza receptors (Matrosovich et al., 2009). We identified 3,862 unique nucleotide sequences encoding 1,458 distinct binding peptide protein mutants. For the "evasion peptide" (Ca1 epitope) case, only human-infecting H1 hemagglutinin nucleotide sequences collected between year 2001 and year 2024 were used. We identified 497 unique nucleotide sequences encoding 181 distinct protein sequences located at the Ca1 antigenic site. In our framework, we designate the nucleotide datasets as the observed nucleotide dataset $\mathcal{D}_\mathcal{N}$ used to compute the emergence probability presented the model proposed in Section 2.4 as a part of the approximated log-likelihood function in Eq. 6. We designate the amino acid datasets translated from the observed nucleotide sequences as the observed amino acid sequence set $\mathcal{D}_n$.

The held-out test dataset for fusion peptides was from studies examining the fusion properties of previously unseen influenza fusion peptide mutants via site-directed mutagenesis (Han et al., 1999; Qiao et al., 1999; Tamm et al., 2002; Lai et al., 2006; Su et al., 2008; Cross et al., 2009). It contains 76 unique amino acid sequences, of which 46 exhibit the fusion property (positive samples) and 30 show impaired fusion (negative samples). Similarly, the test dataset for binding peptides comprises 33 lab-generated mutagenesis results (Yang et al., 2007; Martín et al., 1998; Maines et al., 2011; Chen et al., 2012) and 13 newly observed functional binding peptides from 2025. Among the 46 test sequences, 25 show binding affinity to human influenza receptors, while the remaining 21 sequences show no binding. For the evasion task, the test set contains 51 peptide sequences collected in 2025 and labeled as evasive (functional). To form the non-evasive class, we randomly sampled 51 observed nucleotide sequences and introduced nine mutations to produce unobserved variants, which were then translated to amino acids. With this mutation distance, these sequences are sufficiently dissimilar from the functional set and are unlikely to be evasive, so we treat them as negatives in the test set.

## 3.2 PREDICTING PROTEIN OVERALL FITNESS: PROTEINGYM BENCHMARKS

ProteinGym (Notin et al., 2023) is a large-scale benchmark for protein-fitness prediction, featuring standardized DMS assays and curated clinical datasets with annotated mutation effects. We evaluate Evo-PU on two ProteinGym datasets: (1) PSAE_PICP2 (PSAE) and (2) A0A247D711_LISMN (A0). For each task, Evo-PU is trained on the associated MSA (1,785 sequences for PSAE; 57 for A0) and tested on the corresponding DMS substitution datasets (1,581 sequences for PSAE; 1,653 for A0). Because our evolutionary model operates at the nucleotide level and requires prevalence information–data not available in ProteinGym–we randomly sample a nucleotide sequence encoding each amino-acid MSA sequence and assume equal prevalence across all sequences when running Evo-PU. Although Evo-PU is not designed for general protein-fitness prediction, the goal of this experiment is precisely to assess how well it performs in this broader setting.

## 3.3 EVO-PU: MODEL CHOICES

To demonstrate the flexibility of our framework with different probabilistic classifiers, we evaluate Evo-PU using two model classes: (1) standard logistic regression (LR), and (2) a neural network classifier inspired by the Wide and Deep network architecture (WD) Cheng et al. (2016). We provide the description of the neural network in Appendix B.

To train the Evo-PU classifier, we first construct the set of unobserved nucleotide sequences $\hat{\mathcal{D}}_{\mathcal{N}}$ as outlined in Section 2.4. We approximate the set of previously emerged nucleotide sequences $\mathcal{D}_{\mathcal{N}}^p$ by the observed dataset $\mathcal{D}_{\mathcal{N}}$ and estimate the total count $c(y^p)$ of each $y^p \in \mathcal{D}_{\mathcal{N}}^p$ as $T f(y^p)$, where $f(y^p)$ is the empirical frequency of $y^p$ in $\mathcal{D}_{\mathcal{N}}$ and $T$ is the estimated number of infected hosts in the preceding period. For the influenza datasets, $\mathcal{D}_{\mathcal{N}}$ spans 24 years (2001–2024), we assume the previous step covers the same duration. Using the global estimate of 1 billion influenza cases per year (Nair et al., 2011), we set $T = 24$B. For the ProteinGym experiments, the underlying datasets do not include temporal coverage or prevalence information. To maintain consistency in our implementation of Evo-PU, we therefore adopt the same estimate of $T = 24B$ for these tasks.

From each $y^p \in \mathcal{D}_{\mathcal{N}}^p$, we generate all possible single-nucleotide mutants $y'$, considering both transition and transversion mechanisms (Luo et al., 2016) (See Appendix C for nucleotide mutation patterns). Following prior studies (Wakeley, 1996; Stoltzfus & Norris, 2016; Pauly et al., 2017; Acevedo et al., 2014), we assume mutation probabilities of $\mathbb{P}(y^p \to y') \approx 2.6 \times 10^{-5}$ for transitions and $1.4 \times 10^{-7}$ for transversions.

We then construct the candidate set $\hat{\mathcal{D}}_{\mathcal{N}}'$ of likely emergent but unobserved nucleotide sequences by retaining only those $y'$ satisfying $p_e(y'; \alpha) \geq 1 - \exp(-10)$ under $\alpha = 1$. For the influenza tasks, this procedure yields 30,433; 17,366; and 1,927 unique unobserved nucleotide sequences for the fusion, binding, and evasion peptides, respectively. For the ProteinGym datasets, it yields 302,933 sequences for PSAE and 40,101 for A0. Some of these nucleotide sequences translate into amino acid sequences already present in the data. After removing such duplicates, the remaining sequences give rise to 1,916; 5,203; and 549 unique amino acid sequences for the three influenza tasks, and 167,308 and 26,153 for the PSAE and A0 problems, respectively. These resulting counts determine the cardinality of $\hat{\mathcal{D}}_n'$ used in the likelihood approximation in Eq. 6. The generated nucleotide set

$\hat{\mathcal{D}}'_{\mathcal{N}}$ is used to approximate, for each amino acid sequence $x$, the set of nucleotide sequences that can translate to it, denoted $\hat{\mathcal{Y}}(x)$, which is required to compute the product term in the Evo-PU likelihood.

Directly optimizing the loss in Eq. 7 over discrete amino acid sequences is intractable. To make the optimization feasible, we encode each amino acid using three physicochemical properties that are known to correlate with influenza peptide function (Moon & Fleming, 2011; Foulquier, 2001)–and construct continuous sequence representations by concatenating these encodings. For consistency, we apply the same representation to the ProteinGym problems, although these properties are not tailored to general DMS functional characterization and may therefore limit performance in that setting.

### 3.4 COMPARISON METHODS AND METRIC

We evaluate Evo-PU against several baselines, including the closest PU learning framework for protein design: Protein-PU (Song et al., 2021), a standard PU learning method (2-Step) (Bekker & Davis, 2020). To ensure a fair comparison, we generate unlabeled sets containing 1,865 (for fusion task), 5,535 (for binding task), 461 (for evasion task), 167,308 (for PSAE task) and 26,153 (for A0 task) amino acid sequences—matching the size of $\hat{\mathcal{D}}'_n$ used in Evo-PU. We explore two strategies to generate this unlabeled dataset: (1) random sampling (RAND) and (2) evolutionary knowledge-based generation (E-GEN), where the latter one uses the evolutionary knowledge and generates exactly the same set $\hat{\mathcal{D}}'_n$ as used in our Evo-PU framework. Comparing Evo-PU with PU methods using RAND unlabeled data highlights the benefits of incorporating protein evolution knowledge, while comparisons with those using E-GEN unlabeled data emphasize the advantages of our novel loss function, which integrates the natural selection process.

Moreover, we also compare Evo-PU against the two OCC methods: (1) a standard OC-SVM (Schölkopf et al., 2001) and (2) iForest (Liu et al., 2008). These methods use only the observed set $\mathcal{D}_n$, without incorporating any generated unlabeled data. For consistency, we use the same classifiers (LR or WD) with the same CHEM sequence representation across all PU learning and OCC baselines, matching those used in Evo-PU.

We further benchmark Evo-PU against three DGM-based baselines: (1) the evolutionary model of variant effect (EVE) (Frazer et al., 2021), (2) zero-shot predictions from the protein language model ESM-1v (Meier et al., 2021), and (3) a similarity-based classifier using ESM2 embeddings with k-nearest neighbors (kNN x ESM2) (Esmaili et al., 2025). For kNN x ESM2, we incorporate generated unlabeled data as negative sequences.

All models are evaluated on the same test datasets and compared using the area under the receiver operating characteristic curve (AUC) and average precision (AP), where for both metrics, the higher values indicate better classification performance. A full description of all baseline methods, and optimization details are provided in Appendix D and Appendix E, respectively.

### 3.5 RESULTS AND DISCUSSION

In this section, we report AUC results for two influenza tasks (fusion and binding) and one ProteinGym benchmark (PSAE). For methods involving randomness, we report the mean AUC with error bars. Full results, including AP metrics, are provided in Appendix F. Figure 1 summarizes performance across all methods using LR (top row) and WD classifiers (bottom row). Evo-PU achieves the highest performance in nearly all influenza settings and remains competitive with DGM-based baselines on the ProteinGym benchmarks.

The influenza datasets and ProteinGym benchmarks reflect fundamentally different prediction goals. The influenza tasks focus on a single, well-defined biochemical property (e.g., membrane fusion or receptor binding) of defined local protein motif that is essential for viral survival, while ProteinGym evaluates global protein fitness shaped by multiple biochemical factors simultaneously. Evo-PU is designed to model the data-generating process for one specific property and therefore aligns naturally with the influenza tasks, where observed sequences reflect natural selection acting on that property. Conversely, DGM-based methods-trained to approximate the overall sequence–fitness landscape-are naturally stronger on ProteinGym, a benchmark centered on global fitness prediction. We note that the AUC values of DGM-based methods for the binding task fall below 0.5. We hypothesize that this results from the nature of the test data, which contains lab mutagenesis measurements derived from

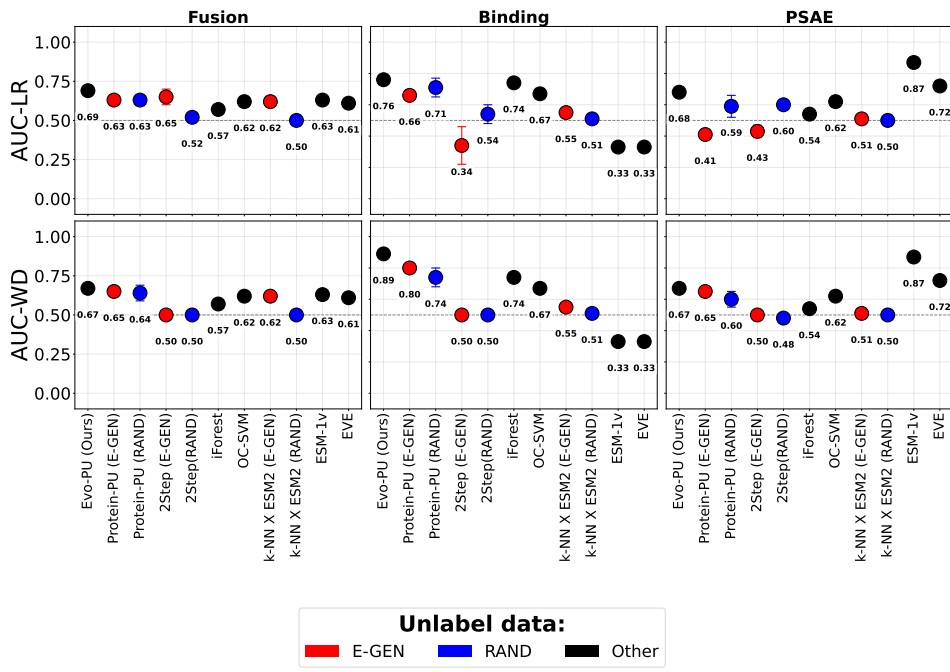

Figure 1: Best average AUC values with error bars across all methods for fusion peptide (left), receptor binding peptide (middle), and PSAE protein (right) classification. Top row: LR classifier with Evo-PU, Protein-PU, and Two-Step. Bottom row: WD classifier with the same methods.

multiple distinct wild-type backgrounds. Such data do not match the assumptions of DGM-based approaches, which typically model variation around a single natural sequence.

For the influenza datasets, Evo-PU consistently surpasses the closest PU-learning baseline (Protein-PU with E-GEN unlabeled data). This improvement comes from two key advantages: 1) Evo-PU uses sequence-dependent class priors that better reflect each sequence's probability of emergence; and 2) it trains a classifier using our novel loss function, which incorporates the effects of natural selection.

Although Evo-PU underperforms DGM-based methods on ProteinGym datasets, this gap may reflect limitations in our protein representation and simplified nucleotide-prevalence assumptions, suggesting opportunities for improvement. Overall, these results show that Evo-PU provides strong predictive performance for tasks involving specific biochemical properties and offers a principled framework for modeling evolutionary accessibility in PU-learning.

## 4 CONCLUSION

We introduced Evo-PU, an evolution-informed positive–unlabeled framework for predicting protein functions critical to organism survival. Evo-PU embeds protein evolution and natural selection at the nucleotide level to assign sequence-dependent class priors and define a biologically grounded likelihood of observing amino acid sequences. As exact likelihood computation is intractable, we use an efficient approximation that focuses on biologically plausible nucleotide-derived variants. We evaluate Evo-PU on three influenza tasks and two ProteinGym benchmarks, where it outperforms state-of-the-art methods on influenza and remains competitive on broader fitness prediction.

While promising, Evo-PU leaves room for improvement. The current model does not consider insertions or deletions, and experimental validation of top predictions would strengthen its practical utility. Extending Evo-PU to high-throughput experimental datasets lacking prevalence information and refining its emergence model offer exciting directions for future work. Additionally, training Evo-PU on human-infecting variants may aid in assessing the avian-to-human transmission risk of emerging avian influenza strains. Together, these avenues highlight Evo-PU's adaptability and potential impact in protein design and biomedical research.

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

# Appendix

## A   LITERATURE REVIEW

In this section, we review existing methods relevant to our Evo-PU framework. We first discuss general approaches, including PU learning, one-class classification (OCC), and deep generative model-based methods, and then highlight specific studies that directly address protein applications, which are most relevant to our work.

**Positive-unlabeled learning:** Typically, PU learning methods involve two primary steps: (1) identifying some unlabeled data as reliable negatives and (2) training a final classifier model using the positive data and reliable negatives. Examples of such methods include Spy-EM (Liu et al. (2002)) and Roc-SVM (Li et al. (2010)). Alternatively, some PU learning methods treat unlabeled data as negative but assign greater importance to positive data by penalizing incorrect predictions of positive instances. Examples include biased-SVM (Liu et al. (2003)) and weighted logistic regression (Lee and Liu (2003)). A particularly relevant PU learning method for our study is the PU learning for protein design (Protein-PU) framework (Song et al. (2021)), which fits a logistic regression model using positive and unlabeled data through a custom loss function that incorporates prior knowledge about the distribution of labeled data.

**One-class classification:** OCC methods can be divided into One-class Support Vector Machine (OSVM)-based and non-OSVM-based approaches (Khan and Madden (2014)). Pioneer OSVM-based methods build a smallest hyper-sphere that encloses positive samples (SVDD) (Tax and Duin (1999a;b; 2001)) or a hyper-plane that separates positive data from the origin (OC-SVM) (Schölkopf et al. (2001)). Recent advances in OSVM-based methods have used neural network for feature extraction and apply traditional OSVM approaches over the extracted features (Erfani et al. (2016); Ghafoori and Leckie (2020)). Examples of non-OSVM-based methods includes the ones using neural network models (Manevitz and Yousef (2001); Skabar (2003); Chalapathy (2018)), decision trees (Liu et al. (2008); Désir et al. (2012); Xu et al. (2023)), nearest neighbors (Munroe and Madden (2005)) and Bayesian classifiers (Wang and Stolfo (2003)). OCC framework has been tailored to protein-related applications. For example, Mei and Zhu (2015) considered a problem of prediciting protein-protein interaction and proposed to use OSVM-based method to sample negative data first and then use the two-class SVM as a final classifier. Yousef and Charkari (2015) proposed to use SVDD together with physicochemical property-based representations of proteins to classify genes with diseases of interest.

**Protein classification using deep generative models:** Recent methods for protein classification leverage deep generative models trained on multiple sequence alignments to capture amino acid distributions and evolutionary conservation. For example, zero-shot prediction via the protein-language-model ESM-1v (Meier et al. (2021)) that computes the fitness likelihood of a queried sequence with respect to a wild type sequence. The Evolutionary Model of Variant Effect (EVE) (Frazer et al. (2021)) predicts pathogenicity by training a variational autoencoder (VAE) on MSA-derived sequences of a human protein of interest. The VAE estimates the relative likelihood of each single amino acid variant compared to the wild type, producing evolutionary indices. These indices are then used to fit a two-component Gaussian mixture model that outputs pathogenicity probabilities. Another example is EVEscape (Thadani et al. (2023)), which extends the EVE framework by combining information from evolutionary scores from EVE with protein structural and chemical information and using logistic functions to predict the likelihood of immune escape in viral variants.

## B   A WIDE AND DEEP NEURAL NETWORK ARCHITECTURE

We customized a neural network structure inspired by (Cheng et al. (2016)) integrates linear memorization with nonlinear generalization for protein function classification. The model takes an input feature vector and the input is processed through two parallel branches: a wide component, consisting of a single fully connected layer that projects the input into a 64-dimensional space, and a deep component, implemented as a two-layer perceptron with 32 and 16 hidden units, each followed by batch normalization, ReLU activation, and dropout ($p$=0.3). The outputs of the wide and deep branches are concatenated into an 80-dimensional joint feature representation, which is then mapped to a

single sigmoid output neuron for binary classification. Weights are initialized with Kaiming-normal initialization.

## C  RNA Nucleotide Mutations

Consider the set of four RNA nucleotides: adenine (A), guanine (G), cytosine (C), and uracil (U). Possible RNA nucleotide mutations via transition and transversion (Luo et al. (2016)) are summarized in Table 1. Specifically, one nucleotide can mutate to another specific nucleotide via transition and two other nucleotides via transversion.

Table 1: Possible scenarios of RNA nucleotide mutations

| RNA Mutations | |
|---|---|
| **Transition** | **Transversion** |
| | (A)→(C) |
| | (A)→(U) |
| (A)→(G) | (G)→(C) |
| (G)→(A) | (G)→(U) |
| (C)→(U) | (C)→(G) |
| (U)→(C) | (C)→(A) |
| | (U)→(A) |
| | (U)→(G) |

## D  Details of Baseline Methods

### D.1  PU-learning methods

**2Step:** In 2Step, 20% of the positive samples are randomly selected and inserted into the unlabeled set as "spies." These spies and the unlabeled data are temporarily treated as negatives, while the remaining 80% of the positives are used as labeled positives. A primary classifier is trained on this combined dataset (spies + unlabeled as negatives, remaining positives as positives). After training, the primary model assigns a probability of being positive to each sequence. The lowest probability among all spy sequences is used as a threshold: any unlabeled sequence with a lower score than this threshold is labeled a reliable negative. The final classifier is then trained using these reliable negatives and all original positives, and is used for final prediction.

**Protein-PU:** Protein-PU (Song et al. (2021)) trains a single classifier on the full positive and unlabeled sets using a custom loss function (Eq. 3) with a constant class prior $q$ as discussed in Section 2.2. Here, we estimate $q$ using the ratio of positive to unlabeled samples, following the original formulation. This yields $q = 0.56, 0.57,$ and $0.58$ for the fusion, binding, and evasion tasks, respectively, and $q = 0.50$ for the both PSAE and A0 ProteinGym problems.

### D.2  OCC methods

For these OCC baselines, we do not incorporate any of the generated sequences. The models are trained using only positive observed sequences for influenza tasks and only provided MSA sequences for ProteinGym benchmarks.

**OC-SVM:** Standard OC-SVM (Schölkopf et al. (2001)) learns a hyperplane separating the traning data from the origin.

**iForest:** iForest (Liu et al. (2008)) scores anomalies based on the number of splits needed to isolate them.

## D.3 Deep generative model-based methods

**EVE**: In EVE (Frazer et al. (2021)), we follow the procedures described in the original paper. For the influenza tasks, we first choose the most frequently observed sequence as the wild type, retrieve similar sequences from the UniRef90 database, construct an MSA, and create the training set by concatenating the relevant MSA segments. For the ProteinGym problems, we directly use the curated MSA datasets provided by the benchmark. We then train a variational autoencoder (VAE) on the one-hot encoded MSA sequences and use it to compute an evolutionary index for each test sequence relative to the wild type. These indices are subsequently modeled with a two-component Gaussian mixture model (GMM) to predict the functional class of each sequence.

**Zero-shot:** For the influenza tasks, we use the most frequently observed sequences as wild-type references and compute the fitness likelihood difference between each test sequence and the wild type using the ESM-1v model, following Eq. 1 in Meier et al. (2021). For the ProteinGym benchmarks, we use the provided wild-type sequences and follow the same likelihood computation.

**Similarity-based method (k-NN x ESM2):** In this baseline, we first embed each protein sequence, including positive sequences, unlabeled generated sequences and test sequences into a latent space via a protein language model ESM2 (Lin et al. (2023)). Then, we train k-nearest neighbor for predictions. We vary the value of $k$ from 2 to 10 and report the one that yields highest AUC. The similar implementation has also been considered for example in Esmaili et al. (2025).

## E  Optimization Details

We implement Evo-PU and all PU-learning baselines in PyTorch (Paszke et al. (2019)) using the Adam optimizer. For Evo-PU, the bounds of $\alpha$ are set to $(0.00075, 1)$ for fusion and two ProteinGym benchmarks, $(0.00025, 1)$ for binding, and $(0.0001, 1)$ for evasion; the bounds of $p_o$ are fixed to $(0.01, 0.99)$ for all tasks. We apply $L_2$ regularization with a penalty of 50, train for 2000 epochs, and use a learning rate of 0.01. For Protein-PU, we tune the learning rate in the range $[10^{-6}, 10^{-2}]$ with 0.1 step size due to its sensitivity, and report the best results.

For Protein-PU with RAND data, we generate 10 unlabeled datasets and report mean AUC and AP and errors across them. For 2Step with E-GEN data, where spy assignment introduces randomness, we report mean AUC, AP, and standard errors across 10 runs. For 2Step with RAND data, we use the same 10 unlabeled datasets as in Protein-PU; for each dataset, we run 10 independent trials with different spy assignments and report mean AUC and AP with error bars across all runs.

For OC-SVM, iForest and k-NN (with ESM2 representation), we use the Scikit-learn implementations (Pedregosa et al. (2011)). The EVE model is run using the official implementation: `https://github.com/OATML-Markslab/EVE`

## F  Full Numerical Results

In this section, we present the full numerical results. The results for influenza tasks (fusion, binding and evasion) are presented in Figures 2 - 4 while the results for ProteinGym (PSAE and A0) are presented in Figure 6 and 5. For each figure, top row presents the results where LR classifier is used and bottom row presents the results were WD classifier is used. The left column reports AUC values while the right column reports AP values.

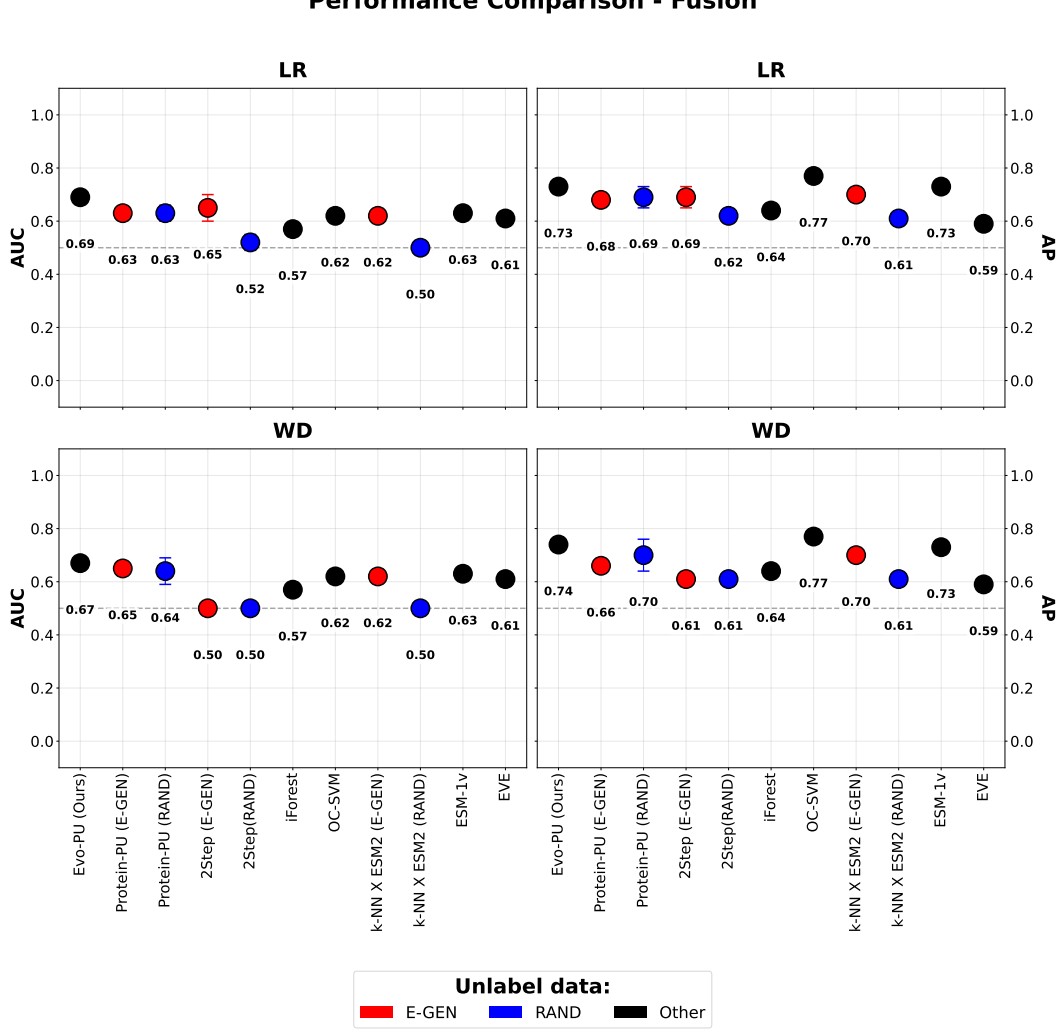

Figure 2: Performance comparison on the fusion tasks. The left column reports AUC values, and the right column reports AP values. The top row shows results using the LR classifier, while the bottom row shows results using the WD classifier.

Figure 3: Performance comparison on the binding tasks. The left column reports AUC values, and the right column reports AP values. The top row shows results using the LR classifier, while the bottom row shows results using the WD classifier.

Figure 4: Performance comparison on the evasion tasks. The left column reports AUC values, and the right column reports AP values. The top row shows results using the LR classifier, while the bottom row shows results using the WD classifier.

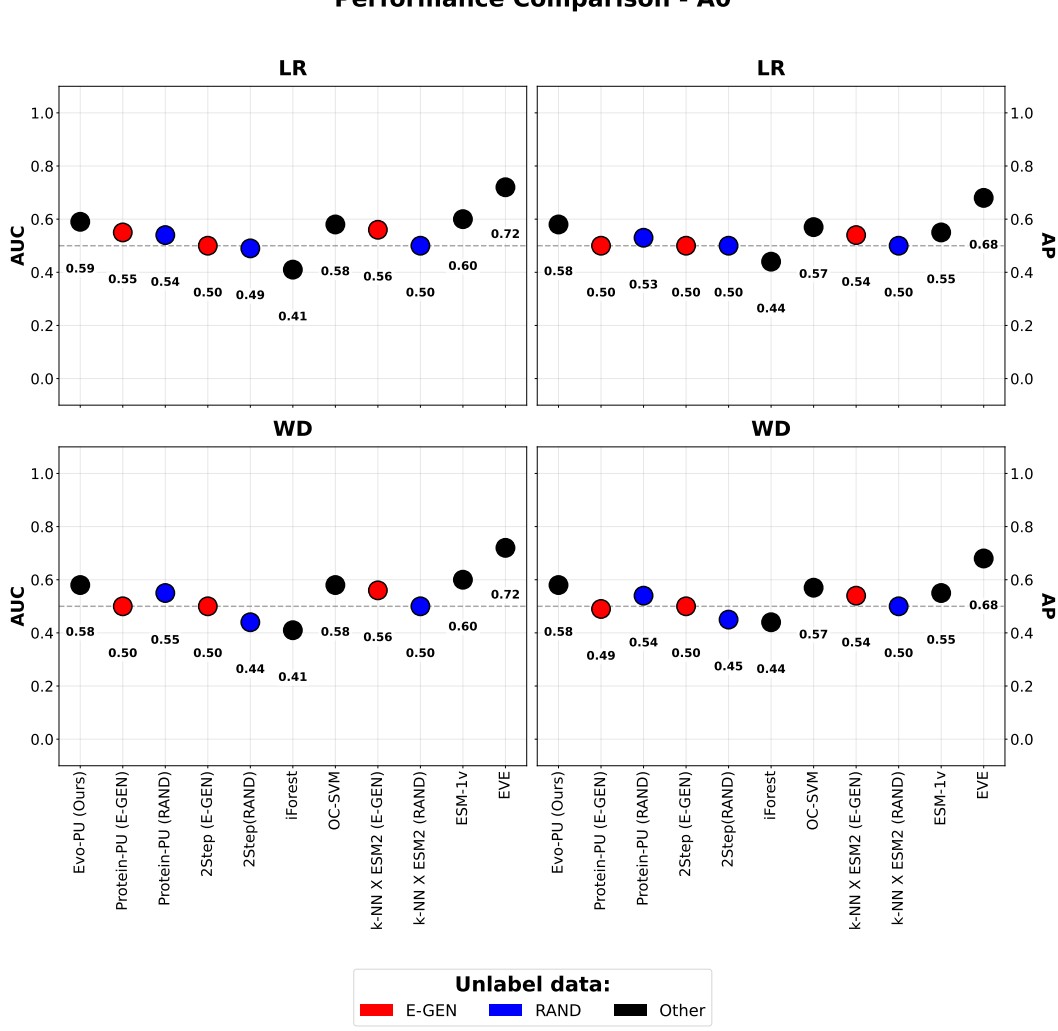

Figure 5: Performance comparison on the ProtienGym-A0. The left column reports AUC values, and the right column reports AP values. The top row shows results using the LR classifier, while the bottom row shows results using the WD classifier.

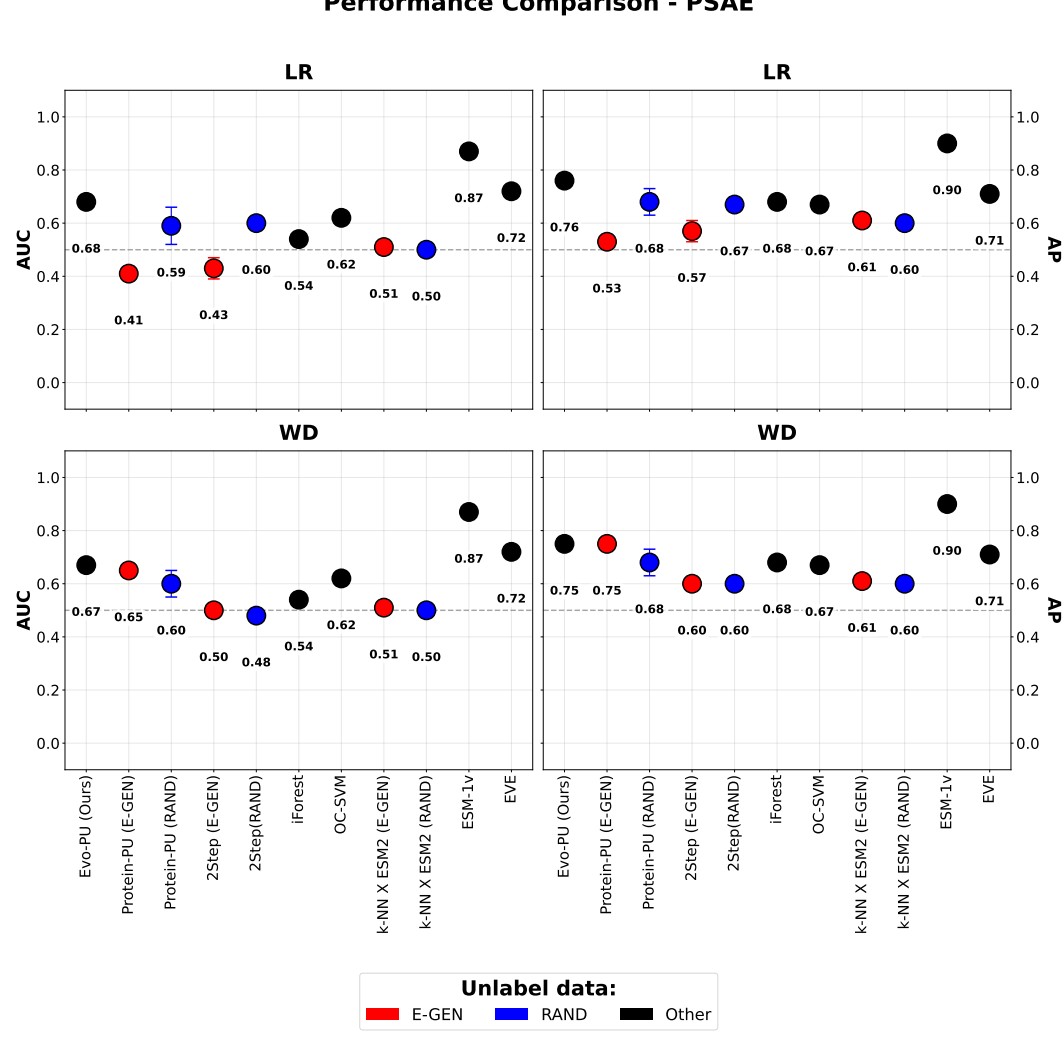

Figure 6: Performance comparison on the ProtienGym-PSAE. The left column reports AUC values, and the right column reports AP values. The top row shows results using the LR classifier, while the bottom row shows results using the WD classifier.

## REFERENCES FOR THE APPENDIX

R Chalapathy. Anomaly detection using one-class neural networks. *arXiv preprint arXiv:1802.06360*, 2018.

Heng-Tze Cheng, Levent Koc, Jeremiah Harmsen, Tal Shaked, Tushar Chandra, Hrishi Aradhye, Glen Anderson, Greg Corrado, Wei Chai, Mustafa Ispir, et al. Wide & deep learning for recommender systems. In *Proceedings of the 1st workshop on deep learning for recommender systems*, pages 7–10, 2016.

Chesner Désir, Simon Bernard, Caroline Petitjean, and Laurent Heutte. A random forest based approach for one class classification in medical imaging. In *Machine Learning in Medical Imaging: Third International Workshop, MLMI 2012, Held in Conjunction with MICCAI 2012, Nice, France, October 1, 2012, Revised Selected Papers 3*, pages 250–257. Springer, 2012.

Sarah M Erfani, Sutharshan Rajasegarar, Shanika Karunasekera, and Christopher Leckie. High-dimensional and large-scale anomaly detection using a linear one-class svm with deep learning. *Pattern Recognition*, 58:121–134, 2016.

Farzaneh Esmaili, Yongfang Qin, Duolin Wang, and Dong Xu. Kinase-substrate prediction using an autoregressive model. *Computational and Structural Biotechnology Journal*, 27:1103–1111, 2025.

Jonathan Frazer, Pascal Notin, Mafalda Dias, Aidan Gomez, Joseph K Min, Kelly Brock, Yarin Gal, and Debora S Marks. Disease variant prediction with deep generative models of evolutionary data. *Nature*, 599(7883):91–95, 2021.

Zahra Ghafoori and Christopher Leckie. Deep multi-sphere support vector data description. In *Proceedings of the 2020 SIAM International Conference on Data Mining*, pages 109–117. SIAM, 2020.

Shehroz S Khan and Michael G Madden. One-class classification: taxonomy of study and review of techniques. *The Knowledge Engineering Review*, 29(3):345–374, 2014.

Wee Sun Lee and Bing Liu. Learning with positive and unlabeled examples using weighted logistic regression. In *ICML*, volume 3, pages 448–455, 2003.

Xiao-Li Li, Bing Liu, and See Kiong Ng. Negative training data can be harmful to text classification. In *Proceedings of the 2010 conference on empirical methods in natural language processing*, pages 218–228, 2010.

Zeming Lin, Halil Akin, Roshan Rao, Brian Hie, Zhongkai Zhu, Wenting Lu, Nikita Smetanin, Robert Verkuil, Ori Kabeli, Yaniv Shmueli, et al. Evolutionary-scale prediction of atomic-level protein structure with a language model. *Science*, 379(6637):1123–1130, 2023.

Bing Liu, Wee Sun Lee, Philip S Yu, and Xiaoli Li. Partially supervised classification of text documents. In *ICML*, volume 2, pages 387–394. Sydney, NSW, 2002.

Bing Liu, Yang Dai, Xiaoli Li, Wee Sun Lee, and Philip S Yu. Building text classifiers using positive and unlabeled examples. In *Third IEEE international conference on data mining*, pages 179–186. IEEE, 2003.

Fei Tony Liu, Kai Ming Ting, and Zhi-Hua Zhou. Isolation forest. In *2008 eighth ieee international conference on data mining*, pages 413–422. IEEE, 2008.

Guang-Hua Luo, Xiao-Huan Li, Zhao-Jun Han, Zhi-Chun Zhang, Qiong Yang, Hui-Fang Guo, and Ji-Chao Fang. Transition and transversion mutations are biased towards gc in transposons of chilo suppressalis (lepidoptera: Pyralidae). *Genes*, 7(10):72, 2016.

Larry M Manevitz and Malik Yousef. One-class svms for document classification. *Journal of machine Learning research*, 2(Dec):139–154, 2001.

Suyu Mei and Hao Zhu. A novel one-class svm based negative data sampling method for recon-structing proteome-wide htlv-human protein interaction networks. *Scientific reports*, 5(1):8034, 2015.

Joshua Meier, Roshan Rao, Robert Verkuil, Jason Liu, Tom Sercu, and Alex Rives. Language models enable zero-shot prediction of the effects of mutations on protein function. *Advances in neural information processing systems*, 34:29287–29303, 2021.

Daniel T Munroe and Michael G Madden. Multi-class and single-class classification approaches to vehicle model recognition from images. *proc. AICS*, pages 1–11, 2005.

Adam Paszke, Sam Gross, Francisco Massa, Adam Lerer, James Bradbury, Gregory Chanan, Trevor Killeen, Zeming Lin, Natalia Gimelshein, Luca Antiga, et al. Pytorch: An imperative style, high-performance deep learning library. *Advances in neural information processing systems*, 32, 2019.

Fabian Pedregosa, Gaël Varoquaux, Alexandre Gramfort, Vincent Michel, Bertrand Thirion, Olivier Grisel, Mathieu Blondel, Peter Prettenhofer, Ron Weiss, Vincent Dubourg, et al. Scikit-learn: Machine learning in python. *the Journal of machine Learning research*, 12:2825–2830, 2011.

Bernhard Schölkopf, John C Platt, John Shawe-Taylor, Alex J Smola, and Robert C Williamson. Estimating the support of a high-dimensional distribution. *Neural computation*, 13(7):1443–1471, 2001.

Andrew Skabar. Single-class classifier learning using neural networks: An application to the prediction of mineral deposits. In *Proceedings of the 2003 International Conference on Machine Learning and Cybernetics (IEEE Cat. No. 03EX693)*, volume 4, pages 2127–2132. IEEE, 2003.

Hyebin Song, Bennett J Bremer, Emily C Hinds, Garvesh Raskutti, and Philip A Romero. Inferring protein sequence-function relationships with large-scale positive-unlabeled learning. *Cell systems*, 12(1):92–101, 2021.

David MJ Tax and Robert PW Duin. Data domain description using support vectors. In *ESANN*, volume 99, pages 251–256, 1999a.

David MJ Tax and Robert PW Duin. Support vector domain description. *Pattern recognition letters*, 20(11-13):1191–1199, 1999b.

David MJ Tax and Robert PW Duin. Uniform object generation for optimizing one-class classifiers. *Journal of machine learning research*, 2(Dec):155–173, 2001.

Nicole N Thadani, Sarah Gurev, Pascal Notin, Noor Youssef, Nathan J Rollins, Daniel Ritter, Chris Sander, Yarin Gal, and Debora S Marks. Learning from prepandemic data to forecast viral escape. *Nature*, 622(7984):818–825, 2023.

Ke Wang and Salvatore Stolfo. One-class training for masquerade detection. 2003.

Hongzuo Xu, Guansong Pang, Yijie Wang, and Yongjun Wang. Deep isolation forest for anomaly detection. *IEEE Transactions on Knowledge and Data Engineering*, 35(12):12591–12604, 2023.

Abdulaziz Yousef and Nasrollah Moghadam Charkari. A novel method based on physicochemical properties of amino acids and one class classification algorithm for disease gene identification. *Journal of biomedical informatics*, 56:300–306, 2015.

