# OpenReview forum: "Evolution-Aware Positive-Unlabeled Learning for Protein Design"
_ICLR.cc/2026/Conference — Submitted to ICLR 2026_

### Official Review · Reviewer_8tkS · 2025-10-17

**Soundness:** 2
**Presentation:** 3
**Contribution:** 2
**Rating:** 4
**Confidence:** 4

**Summary:**

The paper presents EVO-PU, a positive-unlabeled learning framework for classifying favorable versus disruptive protein variants. The problem it addresses is important but not highlighted enough at machine learning conferences, and the proposed method is novel and interesting. Nevertheless, the work is disconnected from modern machine learning approaches.

**Strengths:**

- Data scarcity and the lack of negative data are common challenges in machine learning for biology. This work directly addresses these issues.
- The proposed EVO-PU method is well-explained, novel, and well-justified.
- The results show that EVO-PU outperforms other positive-unlabeled learning and one-class classification approaches.

**Weaknesses:**

The paper's main weakness is its lack of connection to modern machine learning approaches.
- The evaluation is performed on three custom-built datasets based on the Influenza virus. The paper lacks an evaluation on standard benchmarks, such as the well-established ProteinGym [1]. If none of standard datasets used in prior work is suitable, this should be explained.
- The baselines consist mostly of traditional machine learning algorithms. The paper lacks a comparison with modern machine learning models, such as AlphaMissense [2] or even simple ESM2 [3] log-likelihoods [4].
- EVO-PU uses the Wide and Deep (WD) network architecture from 2016 with a simple featurization of protein sequences. The paper would benefit from using a more modern, for example Transformer-based architecture [3]. This would address the issue mentioned on Line 345 (“Directly optimizing the loss in Eq. 7 over discrete amino acid sequences is intractable”), as discrete amino acid sequences can be represented as continuous features [3].

[1] Notin et al., 2023, “ProteinGym: Large-Scale Benchmarks for Protein Design and Fitness Prediction” https://www.biorxiv.org/content/10.1101/2023.12.07.570727v1

[2] Cheng et al., 2023, “Accurate proteome-wide missense variant effect prediction with AlphaMissense” https://www.science.org/doi/10.1126/science.adg7492

[3] Lin et al., 2023, “Evolutionary-scale prediction of atomic-level protein structure with a language model” https://www.science.org/doi/10.1126/science.ade2574

[4] Meier et al, 2021, “Language models enable zero-shot prediction of the effects of mutations on protein function” https://www.biorxiv.org/content/10.1101/2021.07.09.450648v1.full

**Questions:**

1. What is the sequence distance between the training and test data for each of the three datasets?
2. The abstract states, “We consider prediction of protein function, focusing on protein functionalities that enhance survival for one or more organisms.” This suggests that identifying favorable variants is the primary application. If so, would other retrieval-based metrics that account for class imbalance, such as precision and recall, be more informative than the standard AUROC?

---

> ### Author Response · Authors · 2025-11-21
> **Thank you Reviewer 8tkS for your constructive comments**
>
> We are very grateful to the reviewer for their thoughtful, thorough, and well-informed evaluation, particularly the broader ML perspective you bring to our submission. We address each of your comments in detail below.
>
> **C1: Limited evaluation of Evo-PU only on customized influenza dataset and Baseline comparison**
>
> **R1**: Please see the global responses (R1 and R2)
>
> **C2: Transformer-based architecture for discrete amino acid sequence**
>
> **R2**: Thank you for the suggestion. We view the main novelty of our work as the Evo-PU loss function, which can be used to train with essentially any architecture, but we agree that evaluating our loss function on top of a more modern and powerful architecture would better highlight the value of our approach. We are prioritizing the benchmarking tasks suggested by the reviewing team and after this we plan to enhance our approach by training a transformer architecture.with the Evo-PU loss function.
>
>
> **C3: Distance between training and testing sequences**
>
> **R3**: The training and testing sequences differ by only a few point mutations, except in the evasion task, where all negative test sequences are randomly generated and differ by more than nine point mutations from both the training set and the positive test sequences.
>
> **C4: Other evaluation metric**
>
> **R4**: We used AUROC because, although our training data contains only positive sequences, the test sets contain approximately balanced numbers of positive and negative samples, making AUROC an appropriate metric. That said, we agree that a metric looking at precision vs. recall would be a valuable addition because future benchmarks might look at test sets that are less balanced. Our method has a configurable threshold and so rather than reporting precision and recall at a specific threshold, we report average precision (the area under the curve of the precision-vs-recall curve). We now provide average precision in the updated results and will ensure that both metrics are included in the final version.

---

### Official Review · Reviewer_JQFg · 2025-10-21

**Soundness:** 1
**Presentation:** 1
**Contribution:** 1
**Rating:** 2
**Confidence:** 4

**Summary:**

The authors are interested in predicting the effects of mutations from observed sequences from evolution. They cite a biophysical model, suggest a modification based on the choice of negatives and infer models based on this model. They Show their model better predict the effects the effects of mutations from an influenza assay.

**Strengths:**

* The authors suggest a modification to a biophysical model
* They some some (inconsistent) improvement on an assay.

**Weaknesses:**

The main weakness of this work is that it does not properly interface with the mainstream methods for learning evolutionary conserved information from protein sequences, that is, generative models trained on sequences seen across life.
* They only cite EVE and EVEscape. What about ESM, Progen, etc...?
* They claim "While generative models effectively capture conserved constraints, they focus on single-point mutations and often struggle to predict activity for sequences with a few mutations from known positives" without citation. On the contrary, according to ProteinGym, these models are also state of the art for multiple mutations as well.
* They only compare to EVE rather than state-of-the art models that do much better on viral sequence inference.

As well, modern models are evaluated on ProteinGym, which contains hundreds of assays each with thousands of measurements. In contrast, the authors only evaluate their model on a single assay with less than 50 measurements.

**Questions:**

* Why did you train EVE on prevalence data rather than evolutionary data from across life, say by running an alignment. This is another confusion of mine -- in principle your model can be used to perform inference
* "However, since the true wild types for our test sequences are unknown, we instead calculate this index for each test sequence against the top 20 most frequently observed sequences in Dn and select the minimum index. These minimum indices are then modeled with a two-component Gaussian mixture model (GMM) used to predict the probability that a test sequence possesses the property of interest." Why is this a fair choice?
* Much simpler biophysical models have been used to justify mainstream generative models [example](https://proceedings.neurips.cc/paper_files/paper/2022/file/247e592848391fe01f153f179c595090-Paper-Conference.pdf). Could you compare your theory to these?

---

> ### Author Response · Authors · 2025-11-21
> **Thank you Reviewer JQFg for your constructive comments**
>
> We sincerely thank the reviewer for their careful and constructive feedback, and we address each of the points below in detail.
>
> **C1: Baseline comparison and Evo-PU performance on multiple dataset**
>
> **R1**: Please see our global responses (R1 and R2).
>
> **C2**: Implementation of EVE
>
> **R2**: Thank you for this comment. Our initial EVE implementation used an MSA built from our own database so that all methods trained on the same data, but we agree this differs from the original setup. We will revise this by selecting the most frequent sequence as the wild type, constructing an MSA using the broader evolutionary database from the EVE paper, re-running EVE, and reporting updated results.
>
> Regarding our use of the minimum evolutionary index across multiple wild types: all wild types are observed functional sequences, so a low index indicates that a query sequence lies on a plausible evolutionary path from at least one of them, hence likely preserving -- or even improving -- the associated property. We will also evaluate the original single-wild-type formulation and update the results accordingly.
>
> **C3: Comparison to simpler biophysical models:**
>
> **R3**: Thank you very much for sharing the paper. We will read it carefully and provide more comparison in a future response.
>
> **C4: Citations for PLMs-based methods**
>
> **R4**: We will provide citations for these methods in our revised manuscript.

---

### Official Review · Reviewer_HAMc · 2025-11-01

**Soundness:** 1
**Presentation:** 2
**Contribution:** 1
**Rating:** 0
**Confidence:** 5

**Summary:**

This paper introduces Evo-PU, designed for protein binary function prediction where only positive examples are available. The core methodological novelty is the introduction of a sequence-dependent, "evolution-aware" class prior. This evolutionary prior is integrated into a custom likelihood function to train a classifier. The authors evaluate Evo-PU on three tasks related to influenza virus proteins (fusion, binding, and evasion), comparing it against other PU learning methods, one-class classifiers, and an evolution-based generative model (EVE). They report state-of-the-art performance, which they attribute to their more biologically realistic learning framework.

**Strengths:**

1. The paper's core assumption that the sampling bias in natural sequence data is not uniform and can be modeled by evolutionary preference is a strong and valid argument.

2. Strong Ablation (E-GEN vs. RAND): The comparison of baselines using both E-GEN and RAND unlabeled data is a strong control experiment. It successfully suggests that the performance gain of Evo-PU is likely attributable to its unique loss function, not just the quality of the generated unlabeled sequences.

**Weaknesses:**

1. Critically Incomplete and Potentially Misleading Baselines: The experimental evaluation is fundamentally flawed and ignore the most basic and powerful baselines.

      a. The paper fails to compare against standard similarity-based methods, such as a simple k-NN classifier on either BLAST scores, ESM-2 embeddings, or FoldSeek/SaProt structural alignments. These are the fast, robust, and established first-line approaches for function prediction.

     b. The comparison against EVE is conceptually questionable. EVE is a generative model rather than classification model. Furthermore, the reported AUC scores of < 0.5 is flawed.


2. Novelty is Limited and Overstated: The paper's novelty hinges on its "evolution-aware" component. However, this is philosophically very similar to the direct use of scores from pre-trained protein language models (like ESM, AlphaMisense, EVE) to estimate a sequence's plausibility. While the authors integrate this concept into the PU framework in a novel way, the underlying idea is not new. The work feels like a clever recombination of existing ideas (PU learning + evolutionary models) without biological applications and working scenarios.

3. Lack of Connection to Practical Utility: It never demonstrates that its improved AUC score on this specific binary task leads to any tangible downstream benefit. A successful paper would show that its model, for example, can be used to propose a novel immune-evasive viral epitope that is later validated, or that its functional predictions correlate well with clinically relevant outcomes like disease severity. As it stands, the work is an isolated academic exercise in improving a specific metric.

**Questions:**

See weakness above

---

> ### Author Response · Authors · 2025-11-21
> **Thank you Reviewer HAMc for your constructive comments**
>
> We sincerely thank the reviewer for their thoughtful and detailed evaluation. Your comments, particularly regarding missing baselines, novelty framing, and practical relevance, are very valuable and clearly highlight important ways to strengthen our work. We fully agree with these concerns and appreciate the opportunity to address each point below.
>
> **C1: Incomplete and Misleading Baselines**
>
> **R1**: Please see the global responses: R1 and R2.
>
> **C2: Novelty of the Method and Connections to PLM-Based Approaches**
>
> **R2**:
> The main novelty of our work is that we model why protein sequences are observed or missing, not just what the observed sequences look like. PLMs learn patterns only from sequences that already appear in databases, but they do not model the process that determines whether a sequence appears in the data in the first place. We believe that the mechanism behind sequence appearance – how sequences emerge through mutation and how they are captured by surveillance systems – provides strong and underutilized evidence about sequence-function relationships. In Evo-PU, we create a novel observation likelihood function to learn such sequence–function relationships based on the assumption that sequences that are easy to emerge yet remain unobserved are less likely to be functional. We further model the emergence likelihood of each unobserved variant by integrating biochemical knowledge of nucleotide mutation processes.
>
> **C3: Lack of Connection to Practical Utility**
>
> **R3**: We appreciate this concern. Although the current manuscript focuses on establishing the methodological foundation and evaluating Evo-PU on influenza tasks, the framework was designed with downstream biological applications in mind. PLMs model suggested by the reviewers typically use DMS experimental results or viral sequences that emerged later in nature as the ground truth to evaluate model’s performance. And in this work we are using the same approach. Experimental characterization of protein mutation functions from site-mutagenesis are used as ground truth for the fusion and binding tasks. Naturally occurring viral sequences are used as test sequences for binding and evasion tasks.
>
> It is not within the scope of this manuscript, but we are currently validating the functionality of Evo-PU’s predicted functional fusion peptides using experimental approaches. We believe that experimentally validating Evo-PU’s predictions on unseen hemagglutinin mutations will help identify viral variants of concern and guide the development of therapeutics and vaccines against future emerging strains.
>
> In addition, and as explained in our global response, we plan to evaluate Evo-PU on widely-used benchmark datasets such as ProteinGym to ensure that our method is compared fairly against PLM-based approaches and to establish its performance on standard community benchmarks.

---

### Official Review · Reviewer_GfTM · 2025-11-01

**Soundness:** 3
**Presentation:** 2
**Contribution:** 2
**Rating:** 4
**Confidence:** 3

**Summary:**

The paper proposes Evo-PU, a positive-unlabeled learning framework for detecting functional protein sequences. Evo-PU introduces a sequence-dependent class prior derived from a probabilistic model of evolutionary emergence at the nucleotide level. The model estimates A(x), the probability that a protein sequence is functional, effectively serving as a fitness predictor. Empirically, Evo-PU achieves superior AUC scores compared to Protein-PU and other baselines on three influenza case studies (fusion, binding, and immune evasion).

**Strengths:**

- The paper presents a new and well-developed theoretical framework with clear definitions and explanations.
- The proposed method is shown to outperform directly comparable approaches such as Protein-PU.
- The paper is well written and logically organized.

**Weaknesses:**

- Missing comparison to zero-shot protein language model-based fitness (i.e., A(x)) predictors. The paper does not compare Evo-PU to zero-shot fitness predictors derived from protein language models, which have become standard in the field (for example, the ESM-1v paper https://www.biorxiv.org/content/10.1101/2021.07.09.450648v2 or more recent methods for example from the ProteinGym benchmark https://proteingym.org/benchmarks). A comparison on the same influenza benchmark would clarify whether Evo-PU captures additional biological signal beyond what these models already encode implicitly. Without such a comparison, it is difficult to position Evo-PU relative to current state-of-the-art fitness predictors.

- Limited evaluation. The paper evaluates Evo-PU only on a single custom influenza dataset. While this benchmark is carefully constructed and biologically relevant, it has not been used in previous studies. Evaluating Evo-PU on a well-established dataset such as ProteinGym would enable direct comparison to standard baselines and better demonstrate the method’s generality.

**Questions:**

- Could the authors comment on the computational efficiency of Evo-PU? If my understanding is correct, the model must be retrained for each protein sequence, and the computational complexity scales exponentially with sequence length through D_n.

- The theoretical framework operating with both nucleotide and amino acid sequences does not seem to be well justified but introduces a substantial complexity. Would not it be enough to work only on the level of protein sequneces? For example, observability directly operates with protein sequneces in Section 2.2 but is defined on the level of nucleotide sequences.

---

> ### Author Response · Authors · 2025-11-21
> **Thank you Reviewer GfTM for your constructive comments**
>
> We sincerely thank Reviewer GfTM for their thoughtful and constructive comments. We fully agree with the concerns raised, and we believe these points are highly valuable in strengthening the clarity, generality, and overall quality of our work. While we are still running additional experiments, we would like to briefly update you on our progress.
>
> **C1: Benchmark comparisons and evaluation beyond our influenza dataset**
>
> **R1**: Please see the global responses: R1 and R2.
>
>
> **C2: Computational efficiency**
>
> **R2**: Although the model is retrained for each property, its computational complexity is linear, not exponential, in sequence length. This is because we include only single–point mutations in $D_n’$​, reflecting one evolutionary time step. Multiple mutations can occur biologically, but at a significantly lower rate and can be neglected in this short-term evolutionary context, making the single-mutation assumption both reasonable and computationally efficient.
>
> We are extending Evo-PU to be able to include multiple simultaneous point mutations in $D’_n$ in a computationally tractable way. We are doing this by using stochastic gradient descent to sample mutations. Stochastic gradient descent can sample mult-point mutations but does so in a computationally tractable way by being more likely to sample variants that are more likely to occur.
>
>
> **C3: Justification for using both nucleotide and amino acid levels**
>
> **R3**: We agree that clearer justification improves the manuscript. We use both levels because they capture complementary aspects of evolution: protein mutations and emergence likelihood originate at the nucleotide level, while the biological property of protein is determined at the amino acid level. Integrating both allows Evo-PU to better represent evolutionary mechanisms and improve predictive performance. We are revising the manuscript to clarify this. We view the fact that we model the roles of both fundamental building blocks of protein as a strength of our approach.

---

### Author Response · Authors · 2025-11-21
**Thank you reviewers for comments**

First of all, we would like to express our sincere gratitude to all reviewers for taking the time to carefully read our manuscript and provide such thoughtful and constructive feedback. We truly appreciate the effort and expertise that went into these reviews. Below, we summarize common concerns (C) raised across the reviews and provide our corresponding responses (R) as global comments. We will then address all reviewer-specific comments individually in their respective sections.

**C1: Comparison to additional baselines, including protein language model (PLM)–based methods**

**R1**: We fully agree that incorporating stronger and more widely used baselines would significantly strengthen our work. In response, we have implemented the two PLM-related methods suggested by the reviewers:

- Zero-shot PLM-based method: We compute the log-likelihood of each test sequence relative to the wild-type sequence, following the common zero-shot evaluation protocol. The most frequently observed sequence is chosen as the wild type.

- Similarity-based method: We embed sequences using ESM-2 representations and perform prediction with a k-NN classifier. We vary k=2,…,5 and report the best performance.


We have added the corresponding results to **Table 1** below. We note that the numerical values change slightly across runs due to randomness in the initialization of the optimization procedure; however, the overall conclusions remain unchanged. These additional baselines were evaluated on the fusion dataset at this time being. One potential challenge we envision using PLM-based evaluation on the other two tasks is that binding and evasion datasets are constructed by concatenating multiple sequence segments drawn from activity-annotated studies [1-4].  Although we anticipate that the suggested baselines may not perform well on these tasks, we do not believe this reflects limitations of the baseline methods themselves. Nevertheless, we will carry out these additional evaluations as recommended by the reviewers.

Our results show that Evo-PU achieves the highest AUC with the inclusion of suggested baselines and scores the second best average precision (AP) behind OCSVM.
Finally, regarding EVE, we acknowledge the reviewers’ concern about fairness in comparison. We plan to re-run EVE using the original implementation with MSA-based training and will report those results in a future response.

**Table 1: Performance comparison of our Evo-PU to baseline methods. (a) Current methods that have been reported in the main manuscript. (b) and (c) are additional methods; similarity-based and zero-shot methods respectively.**

(a) Current Methods reported in the manuscript
| Method | Data | AUC (CHEM + LR) | AP (CHEM + LR) | AUC (CHEM + WDN) | AP (CHEM + WDN) |
|--------|------|------------------|---------------|------------------|----------------|
| Evo-PU (Ours) | E-Gen | **0.69** | 0.73 | 0.67 | 0.74 |
| Protein-PU | E-Gen | 0.63 | 0.68 | 0.65 | 0.66 |
| Protein-PU | RAND | 0.61 | 0.67 | 0.57 | 0.65 |
| 2Step | E-Gen | 0.62 | 0.68 | 0.50 | 0.61 |
| 2Step | RAND | 0.52 | 0.61 | 0.50 | 0.61 |
| iForest | — | 0.57 | 0.64 | — | — |
| OC-SVM | — | 0.62 | **0.77** | — | — |

(b) ESM2-based similarity model (k-NN)
| Method | Data | AUC | AP |
|--------|------|-----|----|
| Similarity-based | E-Gen | 0.62 | 0.70 |
| Similarity-based | RAND | 0.50 | 0.61 |

(c) Zero-shot baseline (ESM-1v)

| Method | AUC | AP |
|--------|-----|----|
| Zero-shot ESM-1v | 0.63 | 0.73 |

**C2: Performance of our method on benchmark datasets**

**R2**: We plan to perform additional experiments on subsets of ProteinGym assays. To adapt our method to this setting, we will construct the training set as follows:
- Pick WT protein sequence in ProteinGym assays
- Use MSA data associated with the WT available in ProteinGym database
- Train Evo-PU and benchmark using ESM experimental results
- Compare our performance with suggested PLM models

We will include these extended experiments and report the results in a future update.


**References**

[1] Martín, Javier, et al. Virology, 241(1), 101–111 (1998).

[2] De Vries, Robert P., et al. PLoS Pathogens, 13(6), e1006390 (2017).

[3] Yang, Zhi-Yong, et al. Science, 317(5839), 825–828 (2007).

[4] Sriwilaijaroen, N. and Suzuki, Y., Proceedings of the Japan Academy, Series B, 88(6), 226–249. (2012)

---

### Author Response · Authors · 2025-12-03
**Follow-up response on additional baselines and benchmarks**

### **Follow-up 1: Updated Baselines for the Influenza Datasets**

#### **Experimental details**

We provide an updated summary of our baseline experiments on the three influenza datasets (fusion, binding, and evasion). As previously mentioned, we added two additional baselines:

1. **Zero-shot PLM-based method.**
   For each property, we selected the most frequently observed sequences as the wild-type and applied a zero-shot PLM scoring approach.

2. **Similarity-based method.**
   We represented each protein sequence using ESM2 embeddings and applied standard k-NN classification.
   - Training uses all observed positive (functional) sequences and all generated negative (non-functional) sequences.
   - We considered two sources of negative sequences:
     (i) sequences generated from our evolutionary mechanism (the same set used for training Evo-PU), and
     (ii) randomly generated sequences.
   - Both sets contain the same number of sequences per property.
   - We expanded the range for $k$, now varying $k = 2,\dots,10\$, and report the best-performing value based on AUC.

For **EVE**, we followed the original pipeline:
- use the most frequently observed sequence as the wild type,
- retrieve homologs from UniRef90,
- construct an MSA,
- train a VAE,
- compute evolutionary indices,
- fit a Gaussian Mixture Model for classification.

However, additional details are needed due to the structure of our datasets:
- **Fusion peptides** consist of 23 consecutive hemagglutinin residues, so standard EVE preprocessing applies.
- **Binding and evasion peptides** are concatenations of multiple distant hemagglutinin segments. To address this, we:
  1. generated an MSA using a full-length H3 hemagglutinin wild-type sequence,
  2. extracted the aligned subsequences corresponding to the annotated binding/evasion regions,
  3. concatenated them to form the final sequences used for model training.

---

#### **Results on influenza datasets**

Updated results for fusion, binding, and evasion are shown in Tables 1 (updated), 2, and 3.
- **Evo-PU** consistently outperforms all baselines for **fusion** and **binding**, and is competitive with Protein-PU (RAND), iForest, and OCSVM on **evasion**.
- **EVE** and **zero-shot PLM** perform well on fusion but substantially worse on binding and evasion, highlighting a key limitation of generative AI models trained to predict a single global fitness score: they struggle to capture the functional nuances of short, locally acting peptide motifs.
  - We attribute this to **sequence structure**: fusion is the only dataset with contiguous residues, while binding and evasion involve concatenated segments, making PLM- and MSA-based modeling more challenging.

---

> ### Author Response · Authors · 2025-12-03
> **Table 1 Updates**
>
> **Table 1  (Updated): Performance comparison on “FUSION TASK” of our Evo-PU to baseline methods. (a) Current methods that have been reported in the main manuscript. (b), (c) and (d) are additional methods; similarity-based and zero-shot method, and EVE respectively.**
>
> **(a) Current Methods reported in the manuscript**
> | Method | Data | AUC (CHEM + LR) | AP (CHEM + LR) | AUC (CHEM + WDN) | AP (CHEM + WDN) |
> |--------|------|------------------|---------------|------------------|----------------|
> | Evo-PU (Ours) | E-Gen | **0.69** | 0.73 | 0.67 | **0.74** |
> | Protein-PU | E-Gen | 0.63 | 0.68 | 0.65 | 0.66 |
> | Protein-PU | RAND | 0.63 | 0.69 | 0.64 | 0.70 |
> | 2Step | E-Gen | 0.65 | 0.69 | 0.50 | 0.61 |
> | 2Step | RAND | 0.52 | 0.61 | 0.50 | 0.61 |
> | iForest | — | 0.57 | 0.64 | — | — |
> | OC-SVM | — | 0.62 | **0.77** | — | — |
>
> **(b) ESM2-based similarity model (k-NN)**
> | Method | Data | AUC | AP |
> |--------|------|-----|----|
> | Similarity-based | E-Gen | 0.62 | 0.70 |
> | Similarity-based | RAND | 0.50 | 0.61 |
>
> **(c) Zero-shot baseline (ESM-1v)**
>
> | Method | AUC | AP |
> |--------|-----|----|
> | Zero-shot ESM-1v | 0.63 | 0.73 |
>
> **(d) EVE**
>
> | Method | AUC | AP |
> |--------|-----|----|
> | EVE | 0.61 | 0.59 |

---

> > ### Author Response · Authors · 2025-12-03
> > **Table 2**
> >
> > **Table 2: Performance comparison on “BINDING TASK” of our Evo-PU to baseline methods. (a) Current methods that have been reported in the main manuscript. (b), (c) and (d) are additional methods; similarity-based and zero-shot method, and EVE respectively.**
> >
> > **(a) Current Methods reported in the manuscript**
> > | Method | Data | AUC (CHEM + LR) | AP (CHEM + LR) | AUC (CHEM + WDN) | AP (CHEM + WDN) |
> > |--------|------|------------------|---------------|------------------|----------------|
> > | Evo-PU (Ours) | E-Gen | 0.76 | 0.82 |**0.89** | **0.92** |
> > | Protein-PU | E-Gen | 0.66 | 0.76 | 0.80 | 0.86 |
> > | Protein-PU | RAND | 0.71 | 0.73 | 0.74 | 0.76 |
> > | 2Step | E-Gen | 0.34 | 0.45 | 0.50 | 0.52 |
> > | 2Step | RAND | 0.54 | 0.55 | 0.50 | 0.52 |
> > | iForest | — | 0.74 | 0.81 | — | — |
> > | OC-SVM | — | 0.67 | 0.76 | — | — |
> >
> > **(b) ESM2-based similarity model (k-NN)**
> > | Method | Data | AUC | AP |
> > |--------|------|-----|----|
> > | Similarity-based | E-Gen | 0.55 | 0.57 |
> > | Similarity-based | RAND | 0.51 | 0.53 |
> >
> > **(c) Zero-shot baseline (ESM-1v)**
> >
> > | Method | AUC | AP |
> > |--------|-----|----|
> > | Zero-shot ESM-1v | 0.33 | 0.49 |
> >
> > **(d) EVE**
> >
> > | Method | AUC | AP |
> > |--------|-----|----|
> > | EVE | 0.33 | 0.44 |

---

> > > ### Author Response · Authors · 2025-12-03
> > > **Table 3**
> > >
> > > **Table 3: Performance comparison on “EVASION TASK” of our Evo-PU to baseline methods. (a) Current methods that have been reported in the main manuscript. (b), (c) and (d) are additional methods; similarity-based and zero-shot method, and EVE respectively.**
> > >
> > > **(a) Current Methods reported in the manuscript**
> > > | Method | Data | AUC (CHEM + LR) | AP (CHEM + LR) | AUC (CHEM + WDN) | AP (CHEM + WDN) |
> > > |--------|------|------------------|---------------|------------------|----------------|
> > > | Evo-PU (Ours) | E-Gen | 0.96 | 0.97 |0.96 | 0.97 |
> > > | Protein-PU | E-Gen | 0.82 | 0.69 | 0.85 | 0.72 |
> > > | Protein-PU | RAND | **0.98** | **0.99** | 0.99 | 0.99 |
> > > | 2Step | E-Gen | 0.53 | 0.50 | 0.63 | 0.58 |
> > > | 2Step | RAND | 0.88 | 0.82 | 0.84 | 0.77 |
> > > | iForest | — | **0.98** | **0.99** | — | — |
> > > | OC-SVM | — | **0.98** | **0.99** | — | — |
> > >
> > > **(b) ESM2-based similarity model (k-NN)**
> > > | Method | Data | AUC | AP |
> > > |--------|------|-----|----|
> > > | Similarity-based | E-Gen | 0.83 | 0.78 |
> > > | Similarity-based | RAND | 0.72 | 0.69 |
> > >
> > > **(c) Zero-shot baseline (ESM-1v)**
> > >
> > > | Method | AUC | AP |
> > > |--------|-----|----|
> > > | Zero-shot ESM-1v | 0.40 | 0.42 |
> > >
> > > **(d) EVE**
> > >
> > > | Method | AUC | AP |
> > > |--------|-----|----|
> > > | EVE | 0.59 | 0.50 |

---

> ### Author Response · Authors · 2025-12-03
> **Additional Benchmark Tasks**
>
> ### **Follow-up 2: Additional Datasets**
>
> We thank the reviewers for suggesting the **ProteinGym** benchmark. We selected two datasets:
> - **PSAE_PICP2** (PSAE)
> - **A0A247D711_LISMN** (A0)
>
> We used the associated MSA files provided on the ProteinGym website as training data and the associated DMS files as testing data.
>
> For **Evo-PU**:
> - Our emergence model operates at the nucleotide level. For each amino acid sequence, we randomly selected a synonymous nucleotide sequence, assuming all choices have equal prevalence.
>
> For **CHEM-based methods**:
> - Gaps and unknown amino acids in the MSA were treated identically and assigned a designated placeholder value.
>
> Results for PSAE and A0 are provided in Tables 4 and 5.
>
> ---
>
> #### **Findings on the ProteinGym datasets**
>
> As expected:
> - **EVE** performs strongly on the A0 task while **zero-shot** PLM methods perform very strongly on the PASE task. Not demonstrating best performance, but Evo-PU’s performance was comparable with **zero-shot** PLM methods on the A0 task (slightly lower AUC but higher AP) and was comparable with **EVE** on the PSAE task (slightly lower AUC but higher AP).
> - Nonetheless, **Evo-PU outperforms all other baselines**, demonstrating competitive performance outside its primary design domain.
>
> We hypothesize two main reasons for the relative performance of Evo-PU on these datasets:
>
> **Reason 1: Oversimplified prevalence structure**
>    - To fairly compare the performance of Evo-PU against **EVE** and **zero-shot** PLM methods, we adopted MSA data provided in ProteinGym as training data, to train Evo-PU using the training data, we made a few assumptions:
>
>        1. Randomly choose one nucleotide sequence from all possible sequences that encodes each protein sequences in the MSA;
>        2. Assume equal prevalence for all nucleotide sequences
>
> - Although Evo-PU’s performance is comparable to at least one state-of-the-art model, we acknowledge the limitations of our current assumptions. In the present Evo-PU implementation on the ProteinGym tasks, we do not fully exploit realistic differences in how frequently individual nucleotide sequences occur despite this being a key strength we envisioned for Evo-PU. Our current approach generates one random nucleotide sequence per observed amino acid sequence in the MSA and assumes equal prevalence across these synthetic nucleotide sequences, which oversimplifies the true evolutionary landscape.
> Given the challenges Evo-PU faces on ProteinGym, we believe these limitations can be mitigated by developing an amino acid-level emergence model that circumvents the requirement for observed nucleotide sequences. Starting from observed amino acid (protein) sequences, we can incorporate nucleotide-level evolutionary knowledge to model the likelihood of amino acid mutation at each point, hence generating unobserved amino acid variants with realistic emergence likelihoods for Evo-PU. Such an approach would allow Evo-PU to better capture true evolutionary constraints and represents a promising direction for future development.
>
>  - In the influenza tasks, by contrast, Evo-PU has access to heterogeneous prevalence information. If an unobserved sequence can be derived from many highly prevalent observed sequences yet is never actually observed, Evo-PU learns that this unobserved sequence is likely **non-functional** (i.e., unlikely to be active). This mechanism is not available when all sequences are treated as having equal prevalence, which weakens Evo-PU’s ability to distinguish functional from non-functional variants.
>
> **Reason 2: Representation limitations**
>    - CHEM-based features were chosen due to known correlations with influenza fusion, binding, and evasion properties.
>    - For ProteinGym proteins (longer, higher dimensional, and containing many gaps/unknowns), these representations may be less informative and make learning more difficult.
>
> Despite these limitations, **Evo-PU consistently outperforms or matches Protein-PU** across all ProteinGym experiments, highlighting the benefit of:
> - sequence-dependent class priors, and
> - a likelihood-based classifier that closely matches the natural data-generating process.

---

> > ### Author Response · Authors · 2025-12-03
> > **Table 4**
> >
> > **Table 4: Performance comparison on the “ProteinGym PSAE” of our Evo-PU to baseline methods. (a) Current methods that have been considered in the main manuscript. (b), (c) and (d) are additional methods; similarity-based and zero-shot method, and EVE respectively.**
> >
> > **(a) Current Methods reported in the manuscript**
> > | Method | Data | AUC (CHEM + LR) | AP (CHEM + LR) | AUC (CHEM + WDN) | AP (CHEM + WDN) |
> > |--------|------|------------------|---------------|------------------|----------------|
> > | Evo-PU (Ours) | E-Gen |0.68 |0.76 |0.67 | 0.75 |
> > | Protein-PU | E-Gen | 0.41 | 0.53 | 0.65 | 0.75 |
> > | Protein-PU | RAND | 0.59 | 0.68 | 0.60 | 0.68 |
> > | 2Step | E-Gen | 0.43 | 0.57 | 0.50 | 0.60 |
> > | 2Step | RAND | 0.60 | 0.67 | 0.48 | 0.60 |
> > | iForest | — | 0.54 | 0.68 | — | — |
> > | OC-SVM | — | 0.62 | 0.67 | — | — |
> >
> > **(b) ESM2-based similarity model (k-NN)**
> > | Method | Data | AUC | AP |
> > |--------|------|-----|----|
> > | Similarity-based | E-Gen | 0.51 | 0.61 |
> > | Similarity-based | RAND | 0.50 | 0.60 |
> >
> > **(c) Zero-shot baseline (ESM-1v)**
> >
> > | Method | AUC | AP |
> > |--------|-----|----|
> > | Zero-shot ESM-1v | **0.87** | **0.90** |
> >
> > **(d) EVE**
> >
> > | Method | AUC | AP |
> > |--------|-----|----|
> > | EVE | 0.72 | 0.71 |

---

> > > ### Author Response · Authors · 2025-12-03
> > > **Table 5**
> > >
> > > **Table 5: Performance comparison on the “ProteinGym A0” of our Evo-PU to baseline methods. (a) Current methods that have been considered in the main manuscript. (b), (c) and (d) are additional methods; similarity-based and zero-shot method, and EVE respectively.**
> > >
> > > **(a) Current Methods reported in the manuscript**
> > > | Method | Data | AUC (CHEM + LR) | AP (CHEM + LR) | AUC (CHEM + WDN) | AP (CHEM + WDN) |
> > > |--------|------|------------------|---------------|------------------|----------------|
> > > | Evo-PU (Ours) | E-Gen |0.59 |0.58 |0.58 | 0.58 |
> > > | Protein-PU | E-Gen | 0.55 | 0.50 | 0.50 | 0.49 |
> > > | Protein-PU | RAND | 0.54 | 0.53 | 0.55 | 0.54 |
> > > | 2Step | E-Gen | 0.50 | 0.50 | 0.50 | 0.50 |
> > > | 2Step | RAND | 0.49 | 0.50 | 0.44 | 0.45 |
> > > | iForest | — | 0.41 | 0.44 | — | — |
> > > | OC-SVM | — | 0.58 | 0.57 | — | — |
> > >
> > > **(b) ESM2-based similarity model (k-NN)**
> > > | Method | Data | AUC | AP |
> > > |--------|------|-----|----|
> > > | Similarity-based | E-Gen | 0.56 | 0.54 |
> > > | Similarity-based | RAND | 0.50 | 0.50 |
> > >
> > > **(c) Zero-shot baseline (ESM-1v)**
> > >
> > > | Method | AUC | AP |
> > > |--------|-----|----|
> > > | Zero-shot ESM-1v | 0.60 | 0.55 |
> > >
> > > **(d) EVE**
> > >
> > > | Method | AUC | AP |
> > > |--------|-----|----|
> > > | EVE | **0.72** | **0.68** |

---

### Author Response · Authors · 2025-12-03
**Final Discussion Summary and Appreciation**

As the discussion period concludes, we would like to sincerely thank all reviewers and the AC for their time, careful reading, and constructive feedback. We truly appreciate the thoughtful suggestions that have helped us strengthen the paper. Below is a brief summary of the major points we addressed and the corresponding changes in the revised manuscript and abstract. All substantial updates are highlighted in **blue** in the updated PDF.

---

### **Clarification of Key Distinctions Between Evo-PU and Deep Generative Model (DGM)–Based Methods**
We added a dedicated discussion in **Section 2.3** clarifying the central conceptual differences between Evo-PU and DGM-based approaches.
In short:

- **Prediction goal:** Evo-PU is designed to model *a specific protein property* that directly affects organismal survival, whereas DGM-based methods aim to predict *overall fitness* of a whole protein sequence.
- **Modeling assumptions:** DGM-based approaches infer fitness solely from the distribution of observed amino acid sequences using MSA sequences. In contrast, Evo-PU explicitly models evolution at the *nucleotide* level and captures the data-generating process shaped by natural selection through our likelihood formulation.

---

### **Additional Baselines**
We incorporated two widely used baselines:

1. **Zero-shot PLM-based prediction**, and
2. **Similarity-based classification** using ESM2 embeddings with a k-NN classifier.

Implementation details for both baselines are provided in **Appendix D**.

---

### **Additional Benchmark Tasks**
We further evaluated Evo-PU on protein *overall fitness prediction* using two standard benchmarks from **ProteinGym**.
Descriptions of these datasets and our training setup are now included in **Section 3.2**.

---

### **Additional Metrics**
We added the **average precision (AP)** metric for all methods in **Appendix F**.

---

We hope these updates effectively address the reviewers’ concerns. Once again, we thank the reviewers and AC for their valuable feedback, which has significantly improved the quality and clarity of our work.

---

### Meta-Review · Area_Chair_vxWu · 2026-01-06

**Summary:**

This submission proposes Evo-PU, a positive–unlabeled (PU) learning framework for classifying favorable vs. disruptive (functional vs. non-functional) protein variants when only positive examples are observed, by introducing a sequence-dependent “evolution-aware” class prior derived from an evolutionary emergence model and integrating it into a custom likelihood/loss. Overall, while the initial reviews raised serious concerns about baseline completeness and positioning vs. modern protein language model (PLM) approaches, the rebuttal strengthened the empirical case by adding PLM-related baselines and reporting additional benchmark evaluations.

**Reviewer Concerns:**

1. The evaluation/positioning initially lacked strong modern baselines (especially PLM- and similarity-based), making it hard to judge whether Evo-PU adds value beyond standard protein modeling tools (reviewers HAMc, JQFg, GfTM, 8tkS).
2. The experimental scope was viewed as too narrow/custom (influenza-focused) and needed validation on standard benchmarks to support generality (reviewers GfTM, JQFg, 8tkS).
3. The comparison to EVE was questioned as potentially unfair/mismatched (e.g., training data / WT choice / reported metrics), weakening the credibility of claimed improvements over that baseline (reviewers HAMc, JQFg).
4. Multiple reviewers felt the method/architecture choices were not well aligned with modern ML practice, raising concerns about novelty framing and whether the contribution is substantial relative to contemporary transformer/PLM approaches (reviewers HAMc, 8tkS, JQFg).

**Reviewer Scores:**

The reviewer may increase their score in light of the authors’ substantial new empirical results; however, the AC does not believe any such update would be sufficient to shift the overall recommendation from reject to accept.

---

### Decision · Program_Chairs · 2026-01-26

Reject